# Lidar reveals pre-Hispanic low-density urbanism in the Bolivian Amazon

Heiko Prümers[1 ✉], Carla Jaimes Betancourt[2], José Iriarte[3], Mark Robinson[3] & Martin Schaich[4]

Archaeological remains of agrarian-based, low-density urbananism[1–3] have been reported to exist beneath the tropical forests of Southeast Asia, Sri Lanka and Central America[4–6]. However, beyond some large interconnected settlements in southern Amazonia[7–9], there has been no such evidence for pre-Hispanic Amazonia. Here we present lidar data of sites belonging to the Casarabe culture (around AD 500 to AD 1400)[10–13] in the Llanos de Mojos savannah–forest mosaic, southwest Amazonia, revealing the presence of two remarkably large sites (147 ha and 315 ha) in a dense four-tiered settlement system. The Casarabe culture area, as far as known today, spans approximately 4,500 km², with one of the large settlement sites controlling an area of approximately 500 km². The civic-ceremonial architecture of these large settlement sites includes stepped platforms, on top of which lie U-shaped structures, rectangular platform mounds and conical pyramids (which are up to 22 m tall). The large settlement sites are surrounded by ranked concentric polygonal banks and represent central nodes that are connected to lower-ranked sites by straight, raised causeways that stretch over several kilometres. Massive water-management infrastructure, composed of canals and reservoirs, complete the settlement system in an anthropogenically modified landscape. Our results indicate that the Casarabe-culture settlement pattern represents a type of tropical low-density urbanism that has not previously been described in Amazonia.

During the Late Holocene epoch, pre-Hispanic agriculturalists in the Llanos de Mojos, Bolivia, transformed the most-extensive, seasonally flooded, Amazonian savannahs (120,000 km²—roughly the size of England) into productive agricultural and aquacultural landscapes with an apparent diversity in sociopolitical organization, water-control systems and economic bases[14–17]. The southeast sector of the Llanos de Mojos (our study region) benefits from soils that have advantageous agricultural properties because of the deposition of a mid-Holocene sedimentary lobe that creates a slightly more elevated topography than the surrounding Llanos de Mojos, which in turn, provides base-rich, Andean-derived, well-drained soils[18]. The Casarabe culture developed here between around AD 500 and AD 1400, spreading over an area of 4,500 km² (see 'Chronology' in the Supplementary Information, Supplementary Figs. 3–5 and Supplementary Tables 2–4). Previous remote-sensing and field-reconnaissance analyses have revealed the presence of 189 large monumental sites (locally known as 'lomas'), 273 smaller sites and 957 km of canals and causeways[10,19] (Supplementary Table 1). Excavations and bioarchaeology indicate that monumental sites were not unoccupied ceremonial centres but inhabited throughout the year by agriculturalists who cultivated a diversity of crops, with maize (*Zea mays*) as the primary staple[10–12,20,21], and who met their protein needs by hunting[22] and fishing[23].

Despite these important advances in the archaeology of the Casarabe culture, until now, we knew the extent and details of mounded architecture only from less than a handful of isolated sites (Extended Data Figs. 5a, 7) because of the logistical difficulties of mapping sites in tropical forested settings. As a result, our understanding of the civic-ceremonial architecture of the major sites and the regional organization of the Casarabe-culture settlements has remained poorly understood. To remedy this situation, we conducted airborne laser mapping for six areas (10–85 km²) that have known concentrations of major settlements, totalling 204 km² (Fig. 1).

Lidar (light detection and ranging) documented in detail the two large settlement sites and 24 smaller sites, of which only 15 were previously known to exist. The new data allowed us to define a four-tiered hierarchy classification of sites (Supplementary Table 5) on the basis of (1) the dimensions of human-made base platforms; (2) the elaboration of the civic-ceremonial architecture on top of them; (3) the presence, number and total area enclosed by the outermost polygonal enclosures (Figs. 2, 3 and Extended Data Figs. 1–4); (4) the number of constructed, straight causeways leading to the site (Fig. 3); and (5) the scale of investment in water-management infrastructure, including systems of canals and water reservoirs (see Supplementary Information for a detailed description of the architectural elements and a description of representative sites).

## Large settlement sites

Settlements that, at more than 100 ha in size, exceed most other settlements of the same culture many times over, are a very early and

[1]Deutsches Archäologisches Institut, Kommission für Archäologie Aussereuropäischer Kulturen, Bonn, Germany. [2]Department for the Anthropology of the Americas, University of Bonn, Bonn, Germany. [3]Department of Archaeology, College of Humanities, University of Exeter, Exeter, UK. [4]ArcTron 3D, Surveying Technology & Software Development GmbH, Altenthann, Germany. ✉e-mail: heiko.pruemers@dainst.de

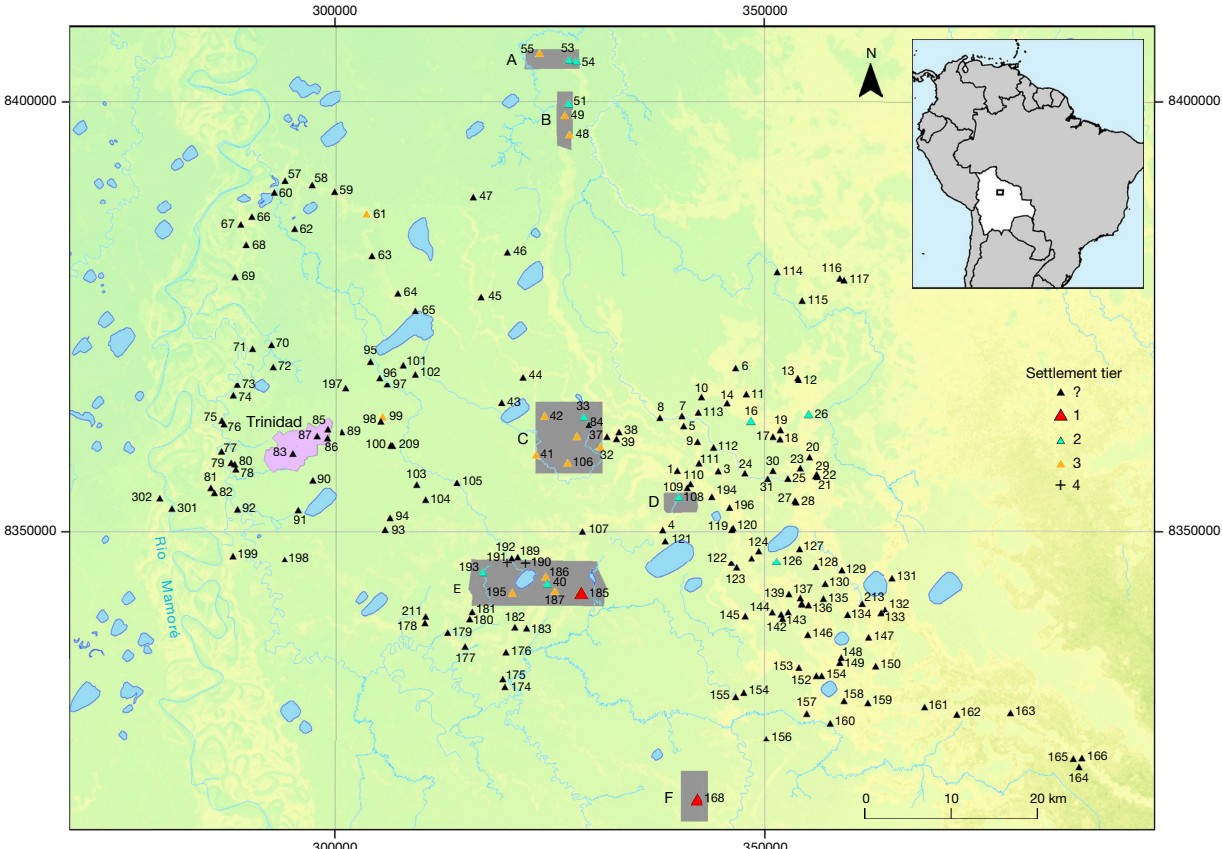

**Fig. 1 | Map of the southeastern Llanos de Mojos.** Lidar coverage is marked by the grey areas (A–F). Black triangles represent settlement sites of the Casarabe culture that have platform mound architecture. The topographical layer is based on TanDEM-X DEM 12-m data.

worldwide phenomenon[2,24,25]. A formally agreed term for these sites is still missing[26], so in this Article we use the descriptive term 'large settlement site' when referring to the two most important sites in the region: Cotoca (147 ha; Figs. 2 and Extended Data Fig. 1) and Landívar (315 ha; Fig. 3 and Extended Data Fig. 2). These two large settlement sites were already known, but their massive size and architectural elaboration became apparent only through the lidar survey.

Both large settlement sites are surrounded by three concentric defensive structures consisting of a moat and rampart (Extended Data Fig. 1), some of which are constituted by double walls (Extended Data Fig. 9a). At the Cotoca site, the inner defensive structures are only preserved in some sections (Extended Data Fig. 1), which may suggest that when the site grew, ramparts were adapted accordingly.

The scale and elaboration of civic-ceremonial architecture are key aspects of the large settlement sites. Massive earthen platform buildings— some of which are U-shaped (Extended Data Fig. 6)—and conical pyramids rise more than 20 m above the surrounding savannah on top of artificial terraces of up to 6 m in height (Fig. 2, Extended Data Figs. 1–3, 4b, 5). The orientation of the buildings that constitute the civic-ceremonial centres of the two large settlement sites is very uniform towards the north-northwest. This probably reflects a cosmological world view, which is also present in the orientation of extended burials of the Casarabe culture (see 'Funerary patterns' in Supplementary Information and Supplementary Fig. 1). The core area of Cotoca (22 ha; Figs. 2, 3b and Extended Data Fig. 1)—as defined by the artificial terrace—is more than three times the size of the secondary centres (Supplementary Table 5). Earth to construct the artificial terraces and platform buildings was acquired, at least partially, from areas adjacent to the settlement's centre. At Cotoca, for example, earth has been removed from a 50- to 80-m wide strip surrounding the central terrace. Today, these lower areas fill with water during the rainy season and are swampy during much of the dry season.

Cotoca and Landívar were the primary centres of a regional settlement network connected by still-visible, straight causeways that radiate from these sites into the landscape for several kilometres (Fig. 3). The presence of platforms (about 20 m by 25 m in size and up to 2 m high) located at strategic points of some of the causeways (Extended Data Fig. 9b, c) and in gaps in the intersection of causeways and polygonal enclosures suggest that access to these large settlement sites may have been restricted and controlled. Labour investment in the construction of the core area of the Landívar site (artificial terraces and platform buildings of the civic-ceremonial centre) is approximately 276,000 m³. The investment in site construction is even more impressive for Cotoca, for which the core area totals 570,690 m³. This latter figure is ten times the amount of earth moved for the construction of the Akapana (53,546 m³)[27], the largest structure in Tiwanaku, capital of the eponymous expansive state that developed over several centuries in the Bolivian highlands, simultaneous with the Casarabe culture.

## Low-density urbanism of the Casarabe culture

The large settlement sites Cotoca and Landívar were primary centres in the settlement network of the Casarabe culture. Secondary centres (El Cerrito and Salvatierra (sites 106, 186, 193 and 195)) (Supplementary Table 1 and Extended Data Figs. 3, 4, 5, 8) are characterized by base platforms ranging from 2 to 6 ha and one still-visible polygonal enclosure that circumscribes areas between 21 and 41 ha. Civic-ceremonial architecture on top of the base platform consists of one to several platform mounds. Tertiary centres (sites 189 and 192) (Extended Data Fig. 8a, b) comprise a base platform of around 0.5 ha with a single platform situated on it and a circular ditch enclosing a maximum area of 2.5 ha. In addition to these built sites, there is a fourth tier composed of a diversity of small (average 0.34 ha) elevated sites (known as forest islands) that

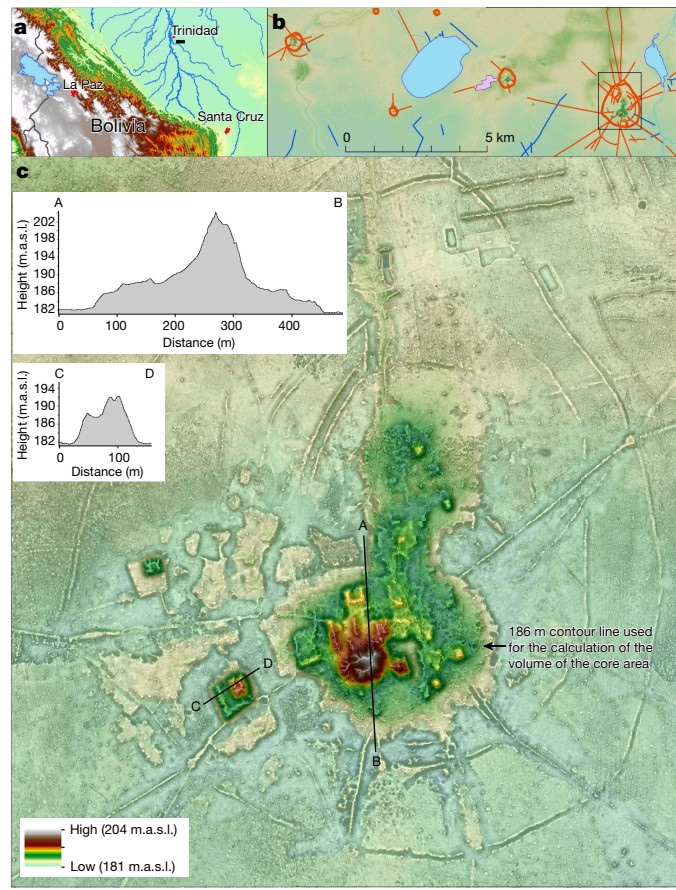

**Fig. 2 | The Cotoca site (no. 185). a**, Occupation of the Cotoca lidar area. **b**, Sites and major archaeological features revealed by lidar in the Cotoca area. **c**, Lidar image of the large settlement site Cotoca with cross sections A–B and C–D. m.a.s.l., metres above sea level.

were probably used as temporary campsites or specialized activity sites. A potential fifth tier of small hamlets possibly exists without mounded architecture that cannot be captured by lidar.

Lidar also reveals a clear correlation between the height of civic-ceremonial architecture and the size of the base platform (Supplementary Fig. 2). Within these broad patterns, the civic-ceremonial architecture within each tier is rather variable. This could be related to

chronology as well as the function of the sites—a matter that will need to be clarified with future studies.

At the regional level, the lidar data combined with previous archaeological-reconnaissance and remote-sensing data show that the Casarabe culture has a highly integrated, continuous and dense settlement system. Across the 4,500 km² with documented Casarabe settlement, there are an average of 10 sites (including primary to tertiary centres) within a 10-km radius of each settlement (that is, within a 2-h walk). Density is higher in the eastern sector, with an average distance of 1,800–3,970 m between settlements (Extended Data Fig. 10). Within this distribution, sites tend to be spatially clustered, interconnected by causeways and canals forming clusters encompassing areas ranging from 100 km² to more than 500 km² (Extended Data Figs. 3b, 8). Lower-tiered sites typically connect to higher-tier sites, with no formal causeways directly connecting the lower-tiered sites to each other. The large settlement Cotoca is the centre of an area of approximately 500 km², half forest and half savannah, which includes 18 other monumental sites, which consist of three secondary centres (sites 186, 193 and 195), two tertiary centres (sites 187, 189 and 192), and clusters of small fourth-tier sites to the southeast and the west (Extended Data Figs. 4a, 8). The central role of Cotoca as a primary site is also underlined by the impressive system of canals and causeways that radiate from the base platform in all of the cardinal directions, connecting with lower-tier sites, the Ibare River to the south, and lakes to the east. A 7-km canal brought water from Laguna San José to Cotoca, indicating the scale of landscape management and labour mobilization. Spatial centrality was not a necessity for the location of the principal sites, with primary sites also appearing on the periphery of site clusters (Fig. 1). In the absence of a primary centre, secondary centres could function as the central node for the lower-tiered sites in the region. For example, the secondary centre El Cerrito (site 33) appears to be the centre of an area of around 100 km² and is surrounded by low-tiered settlements (sites 39, 41 and 106) connected to the core by causeways (Extended Data Fig. 3b).

## Conclusions

Our results put to rest arguments that western Amazonia was sparsely populated in pre-Hispanic times[28]. The architectural layout of large settlement sites of the Casarabe culture indicates that the inhabitants of this region created a new social and public landscape through monumentality. We propose that the Casarabe-culture settlement system is a singular form of tropical agrarian low-density urbanism[2]—to our knowledge, the first known case for the entire tropical lowlands of South

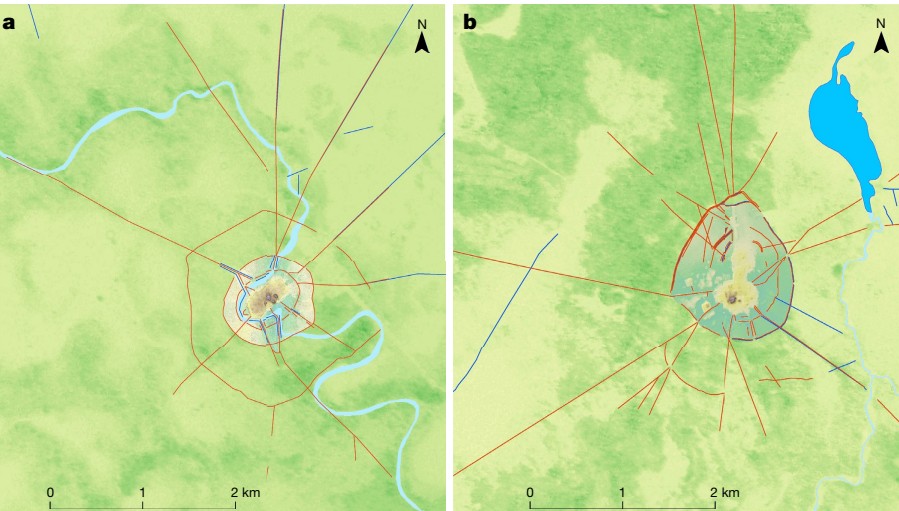

**Fig. 3 | Maps of two large settlement sites.** Red lines indicate the polygonal enclosures and the straight causeways radiating out from the sites. The topographical layers are based on TanDEM-X DEM 12-m data. **a**, Landívar site (no. 168). **b**, Cotoca site (no. 185).

America[29,30]. The scale, monumentality, labour involved in the construction of the civic-ceremonial architecture and water-management infrastructure, and the spatial extent of settlement dispersal compare favourably to Andean cultures and are of a scale far beyond the sophisticated, interconnected settlements of southern Amazonia[31], which lack monumental civic-ceremonial architecture. As such, the data contribute to the discussion of the global wealth of early urban diversity, and will help to redefine the categories used for past and present Amazonian societies.

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

## Methods

### Lidar

Lidar technology has revolutionized the documentation of archaeological sites worldwide, especially those hidden under dense forest[4–6,32–36]. In the Amazon lowlands, lidar was first used for archaeological prospection in the Iténez/Guaporé region as part of the 2011 German–Bolivian Project in Mojos (PABAM)[9]. More recently, lidar has also been used to document archaeological sites in the Amazon regions of Brazil[8] and Peru[37].

In the present study, we mapped a total of 200 km² distributed over 6 unconnected areas of varying size of the Casarabe culture area (Fig. 1). The sensor used was a Riegl VUX-1 scanner, with a Trimble APX-15 UAV GNSS, attached to a Eurocopter AS350 helicopter using a custom mount. The laser pulse repetition rate was 200 kHz. Flight altitude was 200 m above ground level, airspeed was 45 knots. Missions were flown in 200-m parallel strips, with 50% overlap. Data post-processing was done by M.S. (ArcTron) using the RIEGL software RiAnalyze. He successfully overcame fundamental problems in the raw data resulting from a time offset of 18 s between raw laser data and trajectory and unusual height differences between point clouds of adjacent tracks. Through mutual manual and partially automated/iterative corrections, at least visually plausible results were achieved after many attempts. The remaining differences of up to 50 cm had to be accepted. Despite these errors, accuracy in the visualization of the archaeological structures is good. Raw point cloud densities varied between 13 million and 20 million points per km², but generally the density was about 18 million points per km². The filtering was done automatically taking into account, from the outset, only the last pulses and points with only one reflection. The macros created to pre-classify the point cloud were tested using tiles that best reflected the nature of the terrain. Results were then reviewed and modified until an optimal result with only minor residual errors was achieved. At the end of this process digital elevation model (DEM) LAS files were generated that had a mean point spacing of 0.3 m. From these, DEMs with 50 cm per pixel were generated using the 'natural neighbours' method (ArcMap). We used the visualization techniques provided by ArcMap (hillshade, slope) and the relief visualization toolbox (RVT_2.2.1.) developed by the Research Center of the Slovenian Academy of Sciences and Arts[38–40]. Display options were chosen in such a way that they led to an optimal visibility of the archaeological remains.

### Volume calculation

Volumes were calculated for the built structures of the core area (including the base platform with all platforms and truncated pyramids on it) of the three major monumental sites of our study area (Fig. 2 and Extended Data Figs. 2, 3). At El Cerrito, the built base platform covers 61,970 m² and has a volume of 323,988 m³. The base platform of Landívar measures 99,795 m² with a volume of 276,030 m³. The size of the base platform of Cotoca is 159,649 m² and has a volume of 384,228 m³. Volume calculations were done using ArcGIS by (1) extracting the area of the platforms into a new raster; (2) calculating the $\Delta Z$ value of the pixels of that raster; (3) calculating the volume for each pixel; and (4) summing the total volume using the 'zonal statistics as table' command.

### Radiocarbon dating

In total, 144 radiocarbon dates are available for the absolute chronological dating of the Casarabe culture, but these are from a limited number of sites. Stratigraphic information on the dated contexts is available from only four sites (Loma Alta de Casarabe[41,42], Mendoza[12], Salvatierra[11] and Pancho Román[43]. For the two sites investigated by the PABAM, we have 94 radiocarbon dates (46 from Mendoza and 48 from Salvatierra; Supplementary Figs. 3, 4 and Supplementary Tables 2, 3). Two samples were rejected as outliers. Bayesian analysis of the radiocarbon dates was conducted with OxCal v.4.4.2 and the SHcal20 calibration curve. An additional 50 radiocarbon dates have been published for the sites of Los Aceites, Palmasola, Mary, Kiusíu, Loma Alta de Casarabe, Salvatierra and Pancho Román (Supplementary Fig. 5 and Supplementary Table 4).

The 22 radiocarbon dates available for Unit 3 of the Chocolatelito site are unfortunately published in the form of 1α ranges, which prevents their integration into the corresponding list (Supplementary Table 4). Two of the dates fall between AD 0 and 220, which is unusually early and merits closer examination.

### Reporting summary

Further information on research design is available in the Nature Research Reporting Summary linked to this paper.

## Data availability

All relevant data are provided with the paper and its Supplementary Information. The complete datasets used to calibrate all radiocarbon dates are available in Supplementary Tables 2–4.

## Code availability

Code used for the calibration of the ¹⁴C dates in OxCal is available in Supplementary Tables 2–4.

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

**Acknowledgements** We acknowledge the support of the Departmental Autonomous Government of Beni and of the Bolivian Ministry of Cultures and Tourism. We thank E. Méndez P. and G. Nogales H. from Skyplus in Santa Cruz de la Sierra for their patience and excellent work; R. Torrico, Z. Lehm, H. Salas, M. Gonzalez and S. Tin for their support at different stages of this work; R. Landivar for his explanations of the different names by which Loma Landívar is known. This work was funded by the German Archaeological Institute and supported by Intervenciones Urbanas of the Bolivian Planification Ministry. Lidar data were collected with the VUX-1 lidar sensor acquired by the ERC-PAST project (ERC-CoG 616179) to J.I.

**Author contributions** H.P., C.J.B. and J.I. designed the research. H.P., C.J.B., J.I. and M.R. conducted the fieldwork. M.S. processed lidar raw data. H.P. did the geographic information system mapping, statistical analysis and calculations. H.P., C.J.B., J.I. and M.R. carried out the data analysis. H.P., C.J.B., J.I. and M.R. wrote the paper.

**Competing interests** The authors declare no competing interests.

**Additional information**
**Correspondence and requests for materials** should be addressed to Heiko Prümers.

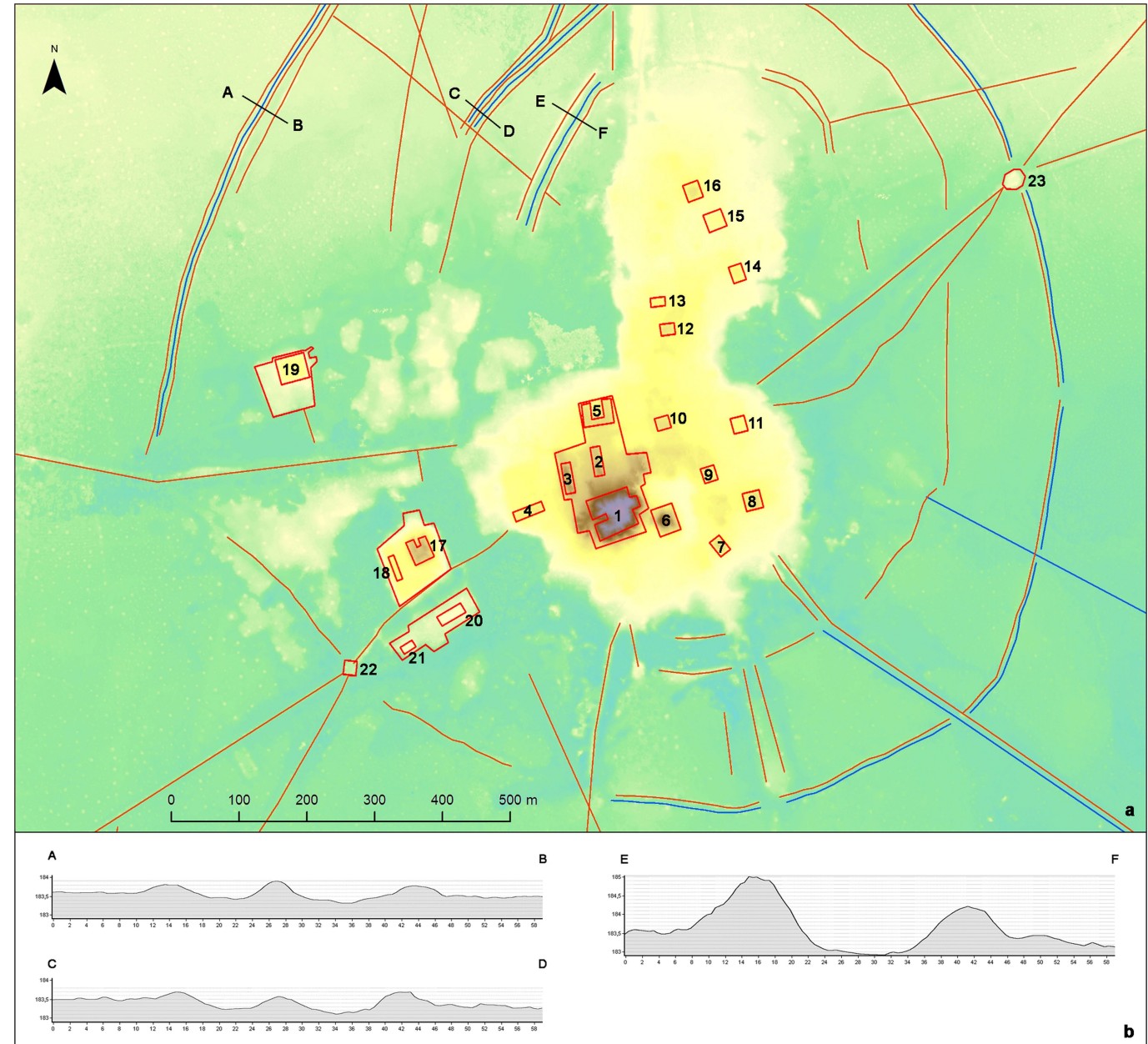

**Extended Data Fig. 1 | Cotoca site (No. 185). a**, Lidar image of the central part of the site. Numbered architectural features: 1, principal, U-shaped mound of 22m height; 2-21 smaller platform mounds; 22 and 23, Platforms at the junction of a causeway and the polygonal enclosure. **b**, Profile cuts of the three enclosures.

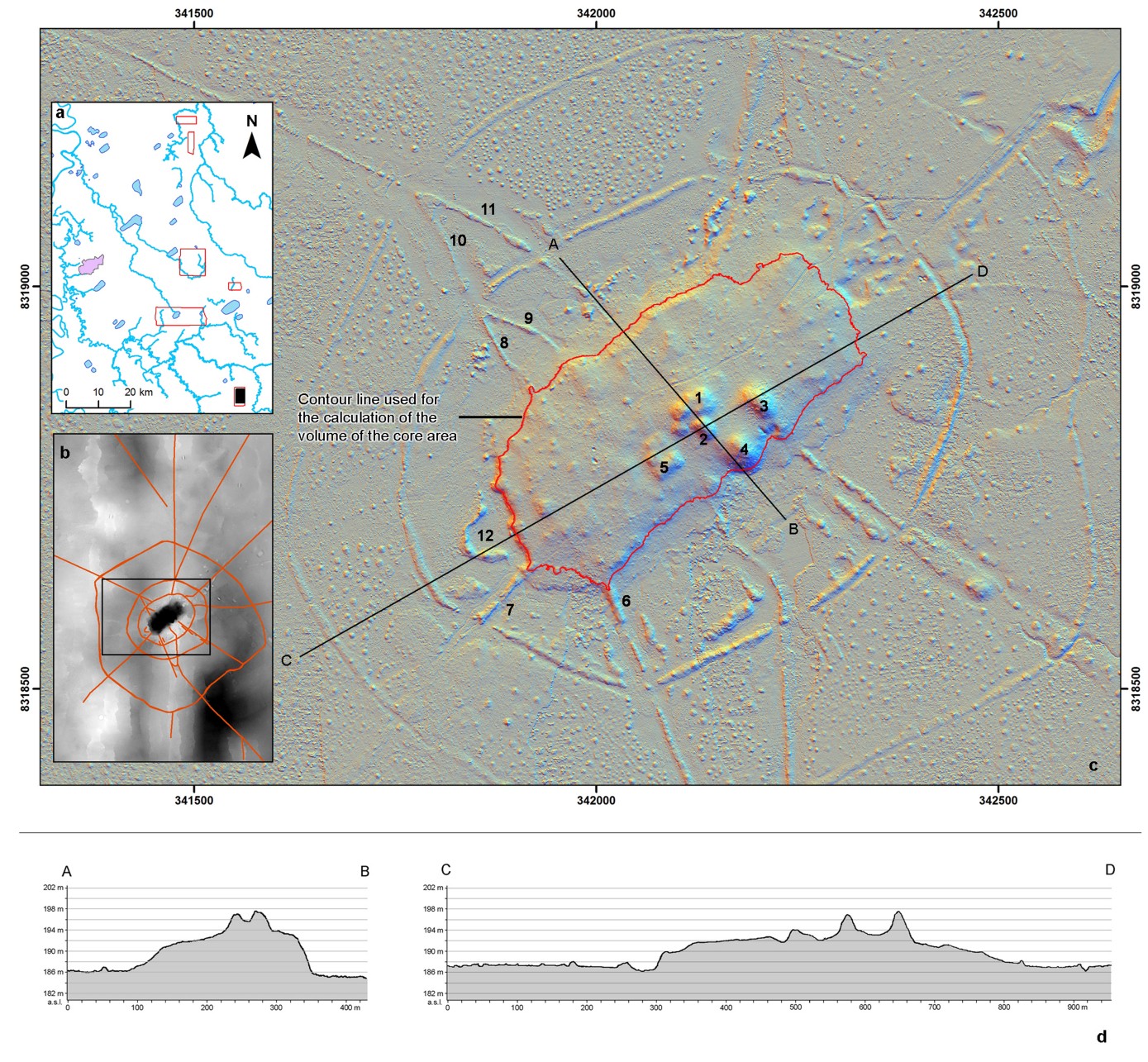

**Extended Data Fig. 2 | Landívar site (No. 168). a**, Location of lidar transect for the Landívar area. **b**, General plan of the site on which the section shown in c is marked. **c**, Lidar image of the with profile cuts. Numbered features: 1-5 platform mounds; 6-11, causeways; 12, structure of unknown function. **d**, Profile cuts A-B and C-D.

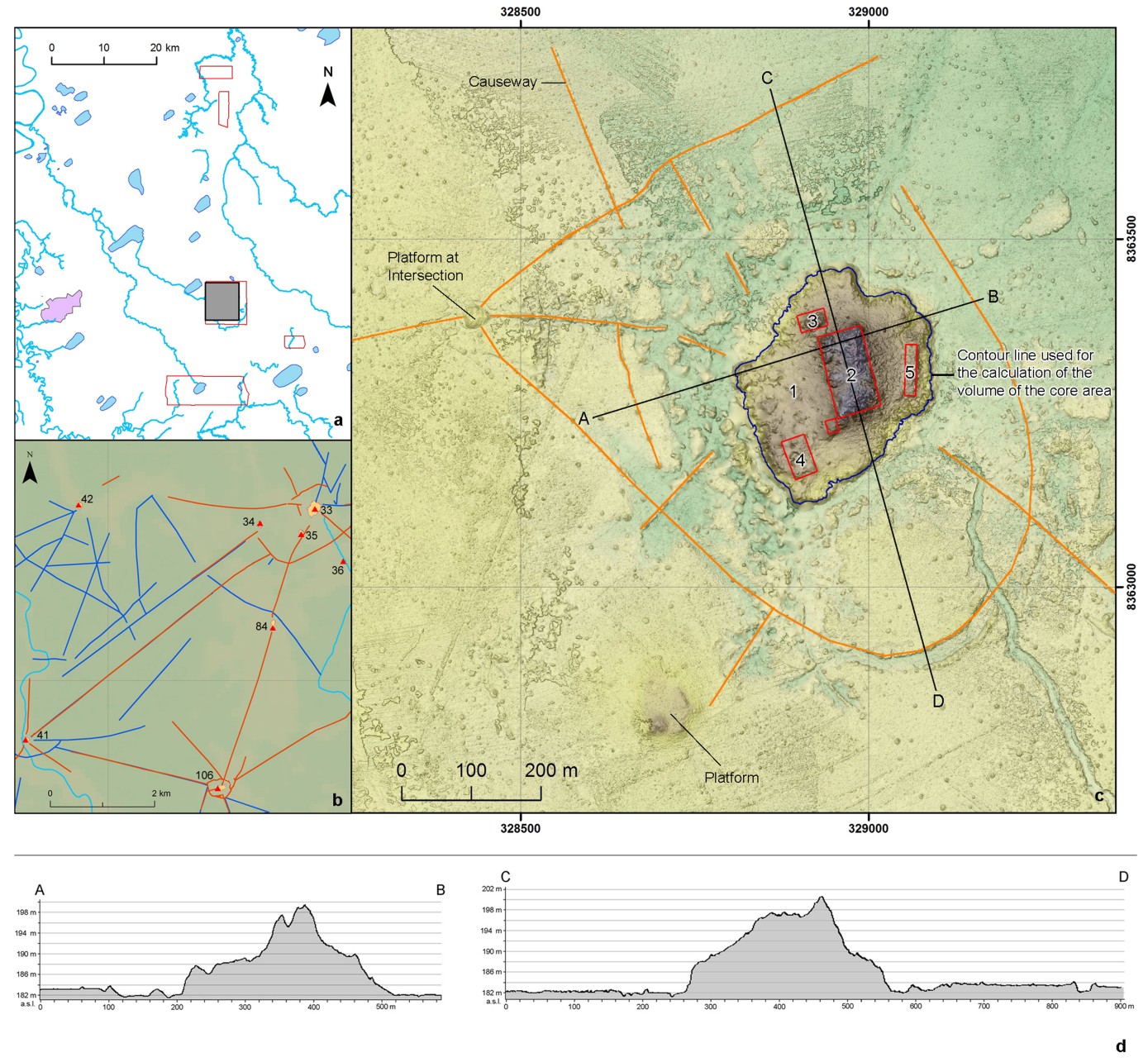

**Extended Data Fig. 3 | El Cerrito site (No. 33). a**, Location of lidar transect for the El Cerrito area. **b**, Canals (blue) and causeways (red) found in the area, some of them connecting the pre-Hispanic sites (red triangles). **c**, Lidar image of El Cerrito. Numbered features: 1, core area platform; 2, truncated pyramid, 3-5, platform mounds. **d**, Profile cuts A-B and C-D.

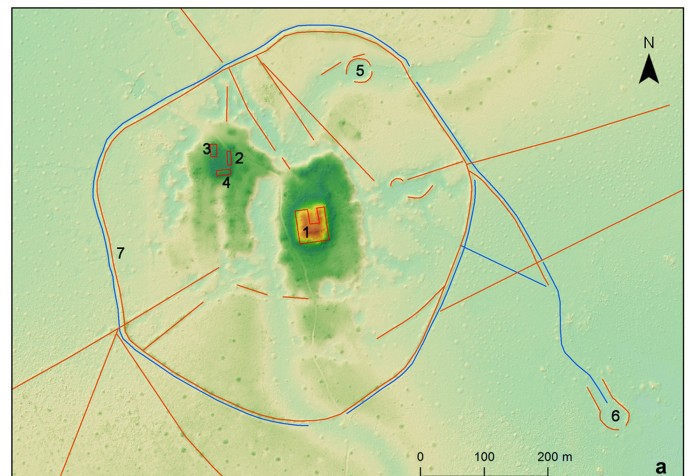
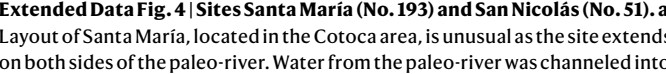
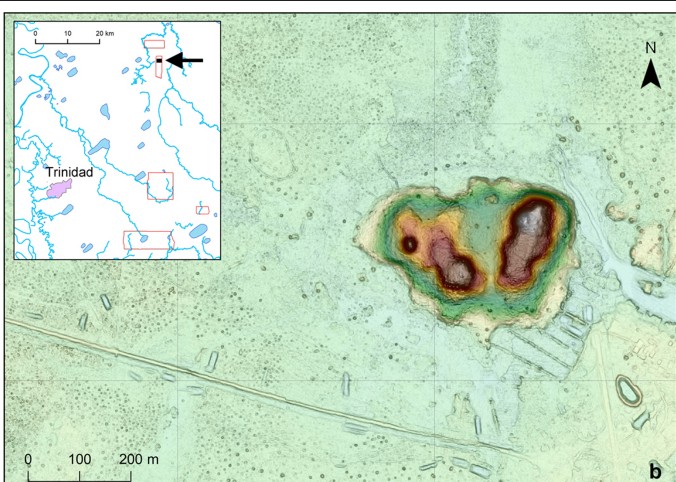

**Extended Data Fig. 4 | Sites Santa María (No. 193) and San Nicolás (No. 51). a**, Layout of Santa María, located in the Cotoca area, is unusual as the site extends on both sides of the paleo-river. Water from the paleo-river was channeled into a circular pond located outside the walled enclosure. Numbered features: 1-4, platform mounds; 5 and 6, reservoir; 7 rampart and moat. **b**, San Nicolás has twin pyramidal structures, one of which is 17 m high.

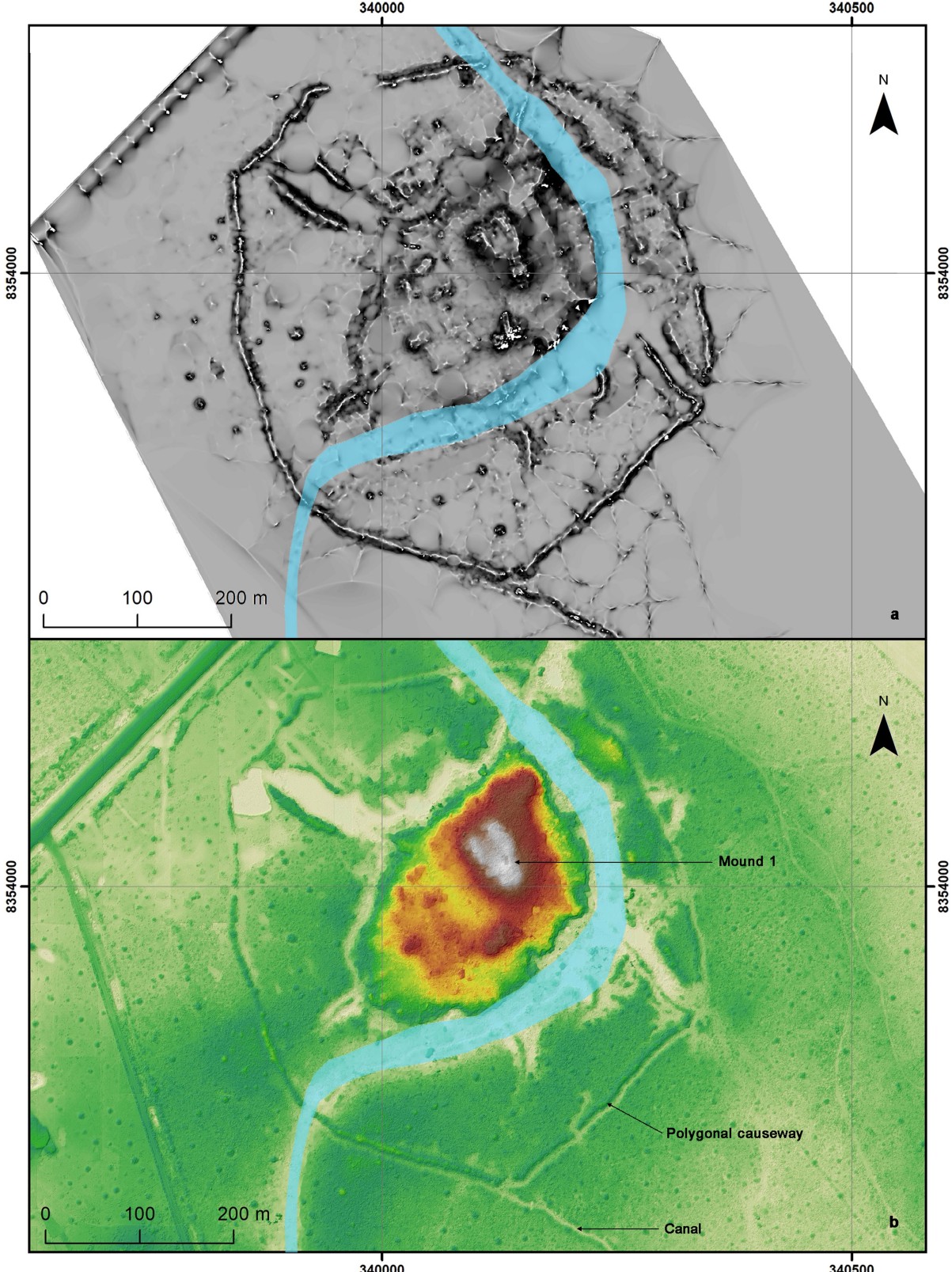

**Extended Data Fig. 5 | Maps of the Salvatierra site (No. 108). a**, DEM calculated from 20.000 points measured with a total-station (Leica TPS 800) during seven months of surveying (campaigns 2004-2006). **b**, High-resolution lidar image (DEM; 50cm/pixel).

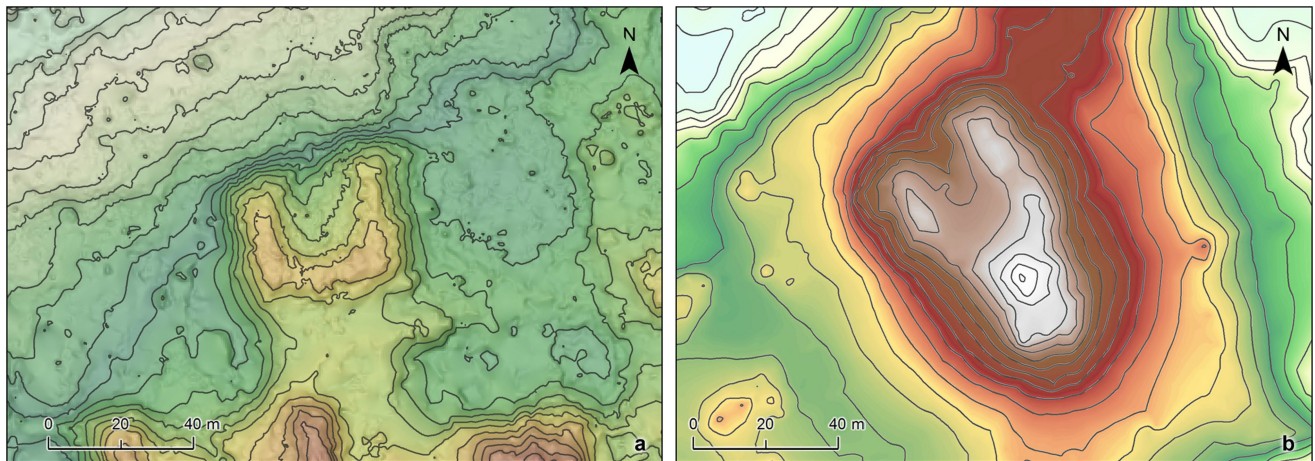

**Extended Data Fig. 6 | Detail maps of two U-shaped platform buildings. a**, one of the lower platform building at the Cotoca site. **b**, principal mound at Salvatierra site. Contour lines: 50 cm.

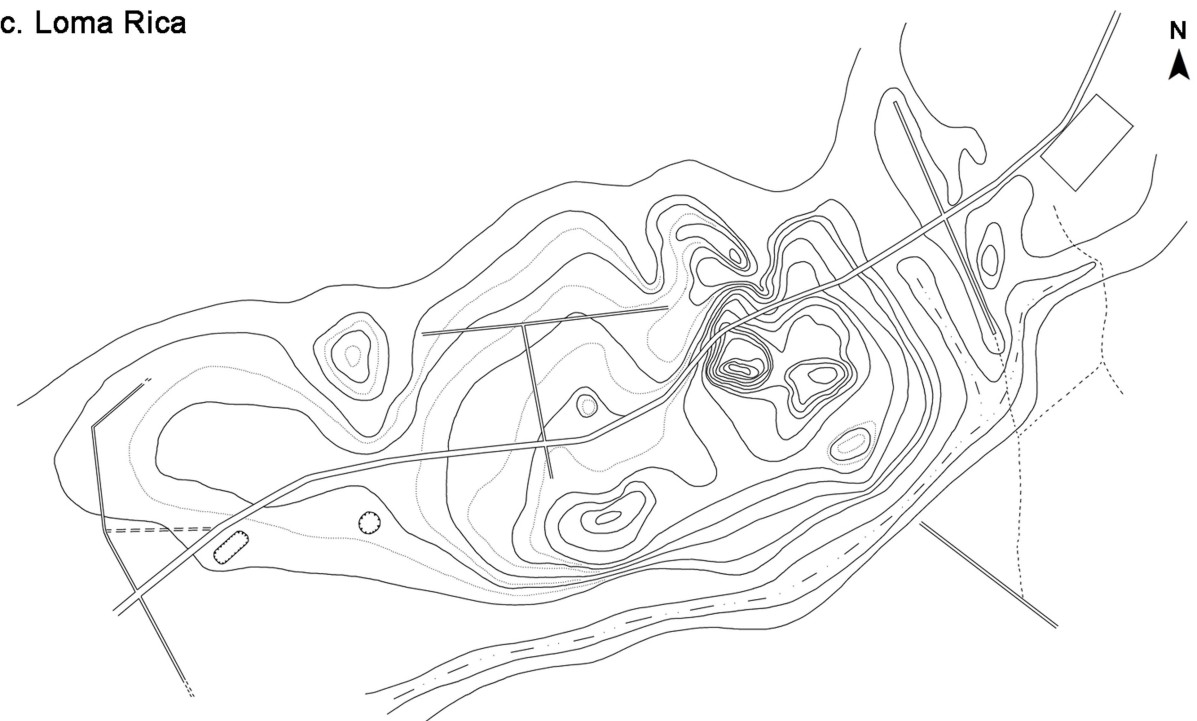

**Extended Data Fig. 7 | Topographic maps of three previously surveyed Casarabe culture sites.** The plans are given in the same scale. **a** and **b** maps of the PABAM-project, redrawn from Prümers 2010: Figs. 29-30[44]; **c** redrawn from Barba 2003: 75, Fig. 5.1[45].

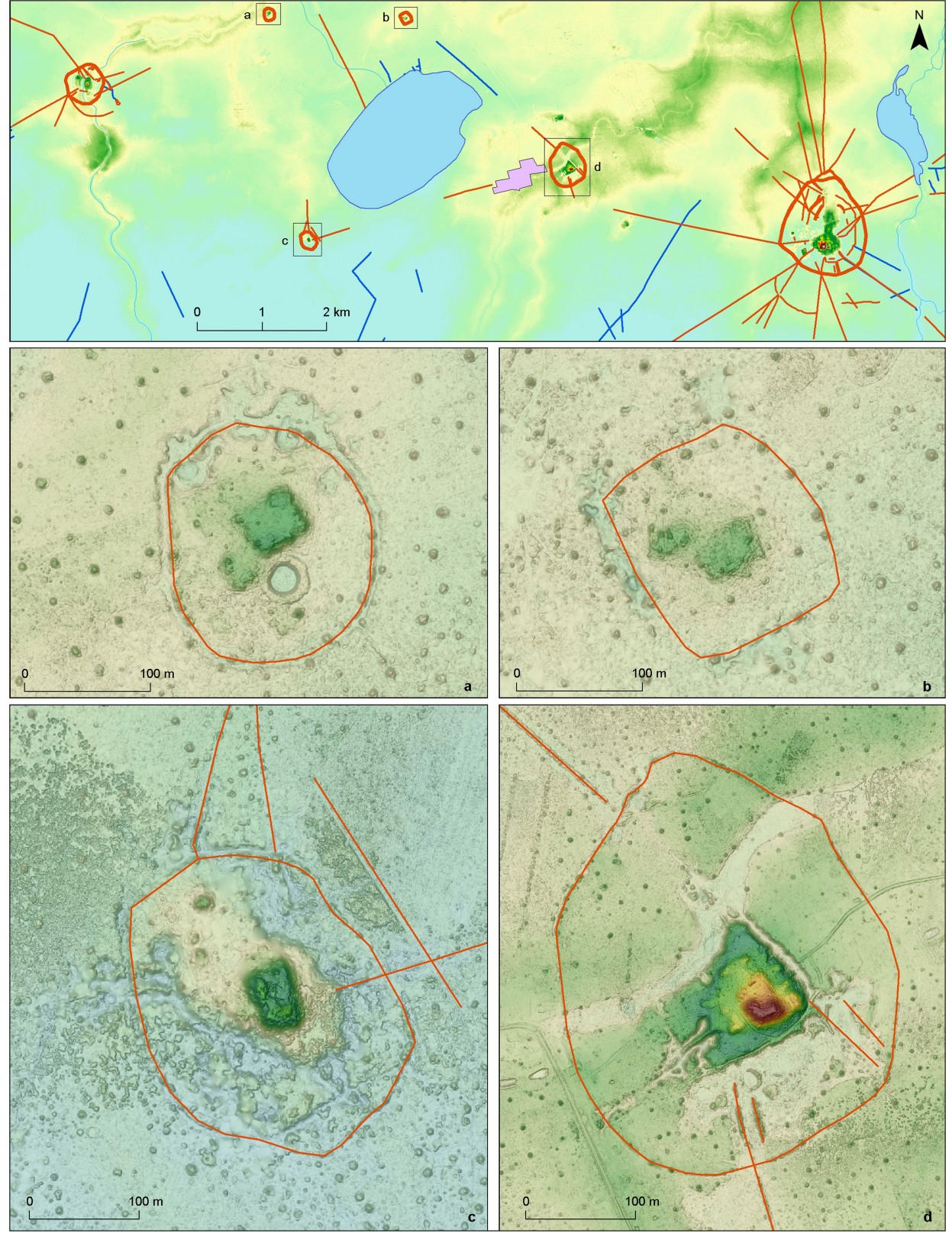

**Extended Data Fig. 8 | Cotoca cluster (see Fig. 1, area E). a**, site 192. **b**, site 189. **c**, site 195. **d**, site 186 (see Supplementary Table 1).

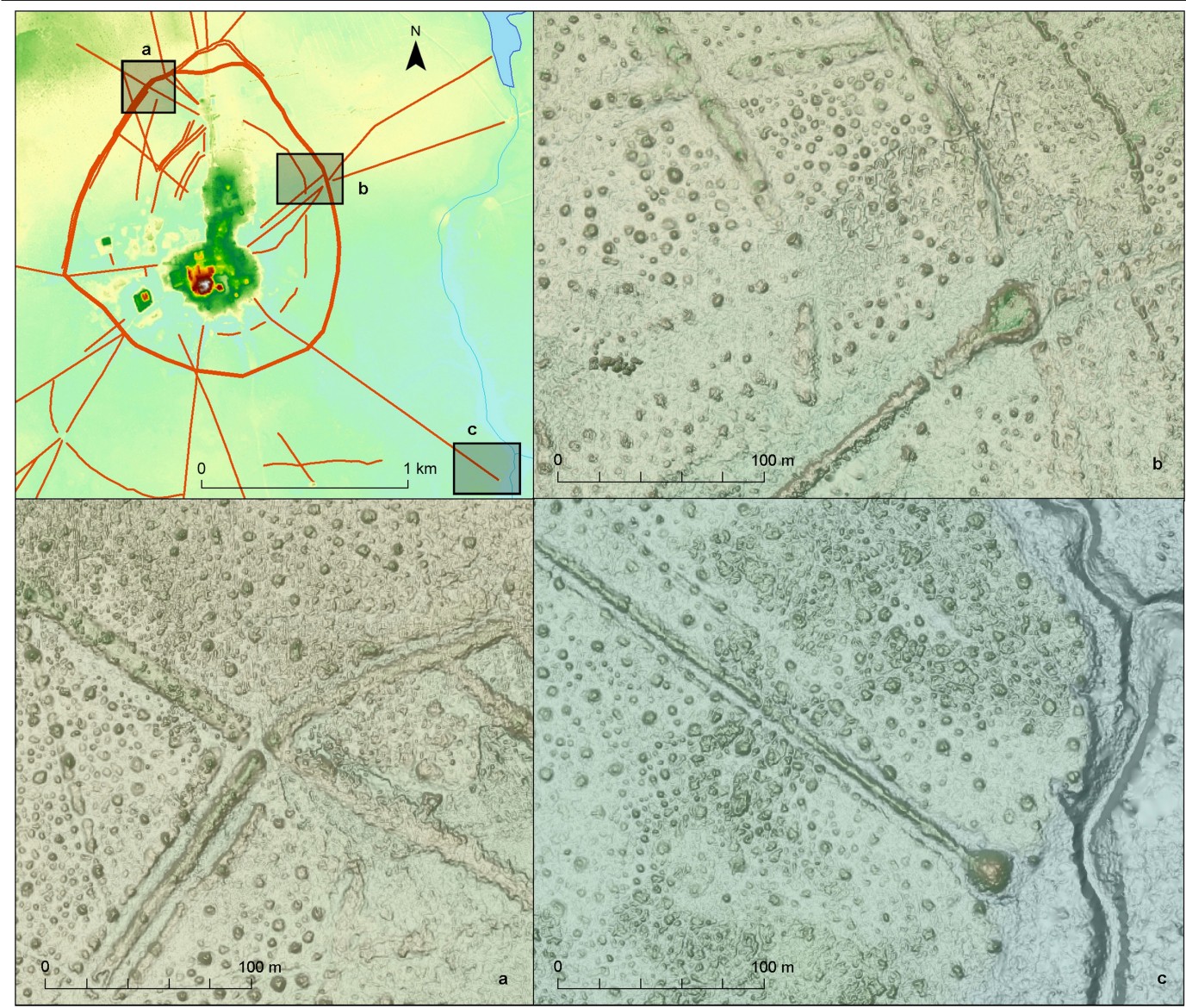

**Extended Data Fig. 9 | Examples of (i) openings in enclosures and straight causeways and (ii) platforms at the intersections of causeways and enclosures at the Cotoca site. a**, Opening at the junction of a causeway and the polygonal enclosure. **b**, Platform at the junction of a causeway and the polygonal enclosure. **c**, Platform located at the opening of the causeway likely caused by a river. On the opposite shore, the causeway continues in a southeastern direction.

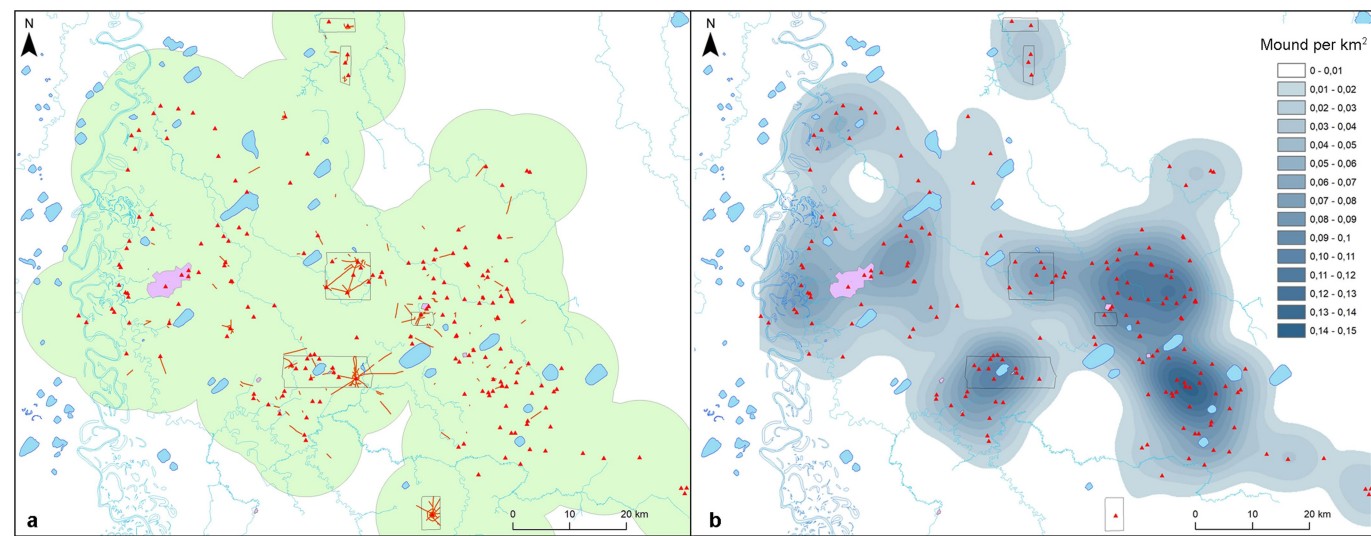

**Extended Data Fig. 10 | Casarabe culture settlement density. a**, 10-km buffer zone map with no buffer boundaries showing how the region was continuously occupied. **b**, Kernel site density map showing the highest concentration of sites in the eastern portion of the study area.

# Reporting Summary

## Statistics

For all statistical analyses, confirm that the following items are present in the figure legend, table legend, main text, or Methods section.

| n/a | Confirmed | |
|---|---|---|
| ☐ | ☒ | The exact sample size (*n*) for each experimental group/condition, given as a discrete number and unit of measurement |
| ☐ | ☒ | A statement on whether measurements were taken from distinct samples or whether the same sample was measured repeatedly |
| ☐ | ☒ | The statistical test(s) used AND whether they are one- or two-sided<br>*Only common tests should be described solely by name; describe more complex techniques in the Methods section.* |
| ☒ | ☐ | A description of all covariates tested |
| ☒ | ☐ | A description of any assumptions or corrections, such as tests of normality and adjustment for multiple comparisons |
| ☐ | ☒ | A full description of the statistical parameters including central tendency (e.g. means) or other basic estimates (e.g. regression coefficient) AND variation (e.g. standard deviation) or associated estimates of uncertainty (e.g. confidence intervals) |
| ☒ | ☐ | For null hypothesis testing, the test statistic (e.g. *F*, *t*, *r*) with confidence intervals, effect sizes, degrees of freedom and *P* value noted<br>*Give P values as exact values whenever suitable.* |
| ☒ | ☐ | For Bayesian analysis, information on the choice of priors and Markov chain Monte Carlo settings |
| ☒ | ☐ | For hierarchical and complex designs, identification of the appropriate level for tests and full reporting of outcomes |
| ☒ | ☐ | Estimates of effect sizes (e.g. Cohen's *d*, Pearson's *r*), indicating how they were calculated |

*Our web collection on statistics for biologists contains articles on many of the points above.*

## Software and code

Policy information about availability of computer code

| | |
|---|---|
| Data collection | RIEGL RiAcquire |
| Data analysis | RIEGL RiAnalyze, ARCGis |

For manuscripts utilizing custom algorithms or software that are central to the research but not yet described in published literature, software must be made available to editors and reviewers. We strongly encourage code deposition in a community repository (e.g. GitHub). See the Nature Portfolio guidelines for submitting code & software for further information.

## Data

Policy information about availability of data

All manuscripts must include a data availability statement. This statement should provide the following information, where applicable:
- Accession codes, unique identifiers, or web links for publicly available datasets
- A description of any restrictions on data availability
- For clinical datasets or third party data, please ensure that the statement adheres to our policy

All relevant data, code, and materials are provided with the paper.

# Field-specific reporting

Please select the one below that is the best fit for your research. If you are not sure, read the appropriate sections before making your selection.

☐ Life sciences ☒ Behavioural & social sciences ☐ Ecological, evolutionary & environmental sciences

For a reference copy of the document with all sections, see nature.com/documents/nr-reporting-summary-flat.pdf

# Behavioural & social sciences study design

All studies must disclose on these points even when the disclosure is negative.

| | |
|---|---|
| Study description | Spatial analysis of lidar data, quantitative |
| Research sample | Laser mapping of six transects (10-85 km2) totalling 204 km2 |
| Sampling strategy | Targeted sampling, areas with know concentrations of archaeological sites |
| Data collection | The sensor used to collect lidar data was a Riegl VUX-1 scanner, with a Trimble APX-15 UAV GNSS, attached to a Eurocopter AS350 helicopter using a custom mount. Laser Pulse Repetition Rate (PRR) was 200 kHz. Flight altitude was 200 m above ground 284 level, airspeed was 45 knots. Missions were flown in 200 m parallel strips, with 50% overlap. |
| Timing | There was no gap between collecting periods. |
| Data exclusions | n/a |
| Non-participation | n/a |
| Randomization | n/a |

# Reporting for specific materials, systems and methods

We require information from authors about some types of materials, experimental systems and methods used in many studies. Here, indicate whether each material, system or method listed is relevant to your study. If you are not sure if a list item applies to your research, read the appropriate section before selecting a response.

## Materials & experimental systems

| n/a | Involved in the study |
|---|---|
| ☒ | ☐ Antibodies |
| ☒ | ☐ Eukaryotic cell lines |
| ☐ | ☒ Palaeontology and archaeology |
| ☒ | ☐ Animals and other organisms |
| ☒ | ☐ Human research participants |
| ☒ | ☐ Clinical data |
| ☒ | ☐ Dual use research of concern |

## Methods

| n/a | Involved in the study |
|---|---|
| ☒ | ☐ ChIP-seq |
| ☒ | ☐ Flow cytometry |
| ☒ | ☐ MRI-based neuroimaging |

## Palaeontology and Archaeology

| | |
|---|---|
| Specimen provenance | Permit to collect lidar data was obtained from the Departmental Autonomous Government of Beni and of the Bolivian Ministry of Cultures and Tourism, Bolivia. |
| Specimen deposition | n/a |
| Dating methods | Radiocarbon essay are reported from previous excavations in the area by the authors and colleagues. |

☒ Tick this box to confirm that the raw and calibrated dates are available in the paper or in Supplementary Information.

| | |
|---|---|
| Ethics oversight | No ethical approval is necessary to conduct lidar mapping in rural areas of Bolivia. |

Note that full information on the approval of the study protocol must also be provided in the manuscript.

