## [Peer Review File · Nature]

Manuscript Title: Lidar reveals pre-Hispanic low-density urbanism in the Bolivian Amazon

Reviewer Comments & Author Rebuttals

Reviewer Reports on the Initial Version:

Referees' comments:

Referee #1 (Remarks to the Author):

This paper reports on new Lidar data for archaeological sites in the Llano de Mojos region of SW Amazonia. Amid a landscape already crowded with known archaeological sites, Lidar data were collected from 6 quadrants (2 large and 4 smaller) that revealed an additional 11 sites (from 15 to 26) and confirmed a 4-tier hierarchy of settlements dendritically connected via straight causeways. These somewhat modest gains are offset by recognition of the massive size of the largest sites (here termed megasites). Use of the term megasite is commendable and preferable over use of the term "city", which invites unresolvable polemics.

The monumental architecture--both U-shaped structures and conical mounds--indicates cultural linkages with the Andes and Pacific coast areas--a topic not pursued by the authors.

Data presentation and supplemental materials are clear and of pretty high quality. Referenced works are appropriate and fairly comprehensive but authors engage in what to my mind is needless hyperbole regarding SW Amazonia. Authors claim that their study represents "a new global case of tropical low-density urbanism and a first for Amazonia" (p. 2). Both Heckenberger and Erickson have presented site evidence from the Amazon that indicates the complexity, density, and greater size of Amazonian sites than previously imagined.

This work builds on those studies and indicates that SW Amazonia was not a sparsely settled backwater but a densely settled and hierarchically organized part of the larger Amazonian world.

Referee #3 (Remarks to the Author):

The results presented here are significant to the interpretation of the archaeological record in eastern Bolivia, but also in the Andes and in the Amazon more generally. My main critique of the manuscript is not with the methods, the documentation, or the conclusions, but that the paper should do more, to make the case that these results have broader implications. The paper is based on a long and impressive record of research in the area. The conclusions of the paper draw on this record, in the form of site locations, excavation results, and radiocarbon dating. The manuscript should be published with revisions. Sharpening the focus of the discussion will give the paper more impact, and a larger audience.

The paper reconstructs a four-tiered settlement hierarchy, and clearly lays out criteria for these four tiers (and a possible fifth). What would make this argument stronger would be to include a map or at least counts of how many sites are included in each category, and how they are arranged on the ground. For example, at Cotoca, one 1st tier site is said to be accompanied by 3 2nd tier sites, 2 3rd tier sites, and (presumably) 13 4th tier sites. More information about the arrangement of these sites and their relationships with one another are needed if the settlement hierarchy is going to support a strong interpretation. Why are the distinctions between the tiers made at those particular points; is the difference between 2nd and 3rd tier really significant? Why is a 2nd interpreted differently from a 3rd? Is the difference between 1 and 2 the same kind of difference as the difference between 3 and 4?

In the past, 4-tier settlement hierarchies have been held to stand for 4-tiered political organizations (like states), is that what the authors are proposing? This is not clear as written, and the paper would be strengthened if it contained a stronger statement about what the pattern represents. Does the spatial hierarchy represent a social hierarchy?

Similarly, the paper uses the term urbanism, but does not make a detailed comparison to the Xingu example (which is cited in the abstract). Is the Casarabe case related to the Xingu case? If so, then how? Can specific comparisons in terms of size of constructions be made? How does this interpretation compare to the interpretations that have been made for the Xingu pattern? Souza et al. compare similar earthworks (or at least comparable earthworks) across the entire Southern Amazon, this work should be mentioned and integrated into the discussion. For megasites (outside Mojos) why is it that those sites are not just called cities? What are the main issues that the introduction alludes to about megasites and cities? How would Casarabe fit into that discussion? Two comparisons are made to the Andean archaeological record, which would be stronger if they were more explicit. The calculation of earth moved to create the Casarabe mounds in comparison to the Akapana is excellent, and could be pursued further: if assumptions are made about what large-scale architecture at Tiwanaku means, can those same assumptions be made about Casarabe? The proposed U-shaped constructions are a very interesting and important parallel to architecture in the highlands and on the coast; this evidence should be documented in greater detail. A skeptical audience might imagine that what appears to be a U-shape on the map is only a mound that has been dug into, or eroded, for example. A little more detail would disarm such criticism. U-shaped constructions appear to go back thousands of years in South America, is the Casarabe phenomenon part of this long-lived tradition?

The authors are unequivocal that this paper settles the question of whether western Amazonia was sparsely settled before 1492, and I agree. I think that this point would hit home for the non-specialist reader with a brief explanation of the potential of the LIDAR methodology in other forested locations. A single sentence about how the LIDAR plots were chosen, and (presumably) the earthworks that were mapped were not visible on other satellite or aerial photography, would then lead the reader to the question of how many more earthworks are present not only in eastern Bolivia (ie in the maps presented here), but in the western Amazon more generally (and perhaps across the entire Amazon). This region is comparatively well studied, after all, and the current paper makes it clear that many earthworks had gone undetected.

The two potential supra-regional interpretations, one pointing toward the Andes and the other pointing towards the Amazon, should both be addressed. Mojos is sometimes in a strange position of being written out of histories of the Andes, because it is clearly not Andean, and also being written out of the Amazon, because it is a unique case, and at least for some people, not really

Amazonian. Neither the Andean audience nor the Amazonian audience should be permitted to safely tuck this data away in a folder marked “anomalous” and therefore stop thinking about it. This research shows that Mojos is significant to both the Andes and the Amazon.

It is a great credit to the research and the manuscript that it leads to these kinds of discussions. Because of space limitations it is might not possible to address all of these larger points. I would advocate for choosing at least one of the implications and pursuing in more detail, and then leaving the others as open questions that can be returned to in future research.

Notes from the manuscript, figures, and supplementary information (by line number)

Title and abstract set the expectation for discussion of megasites, urbanism, u-shaped structures, “commanding” an area of 500 km², massive water management, four-tiered settlement system; a long list of subjects to address.

37—not so much a “global case” as a case that fits into some larger idea?

46—is this meant as a difference from other areas within the LM?

79-80—they may have been known, but not as megasites; this undersells the importance of the research.

84—is there any evidence of palisades in the area, or other corroborating evidence for an interpretation of these earthworks as defensive structures?

86—The presentation of the U-shaped buildings needs more detail because of the interpretive weight that these constructions carry (esp. among Andeanists)

107—comparison to Akapana is excellent, are there any other comparisons (Xingu? Maya? SE Asia) that could be briefly added here to supplement the comparative perspective (which the abstract seems to promise).

112-122—would it be possible to know more about how these classes were selected, why these are significant “break” points?

138—if there are 3 secondary and 2 tertiary, does this have any implications for the interpretation of the four tier hierarchy?

151—this is an excellent point and the paper could even go farther in this direction with more information about how many earthworks were found in the given area, and then discuss how much more could be found not just in eastern Bolivia but in the western Amazon more generally.

Figure 1—strongly makes the point that the LIDAR coverage only includes a fraction of the area; this could be emphasized more strongly in the caption or somewhere in the text.

Figure 2—this figure is excellent; perhaps add a legend for the color scale/elevation. Recognizing the problem of space, is it possible to indicate which mounds or earthworks were unknown? Perhaps in the text or caption? (this is clear in the supplemental information on the list, but maybe it can be shown here)

Figure 3—Good image, this multiplies the impact of Figure 2 because it shows that it is not a unique case.

298—state the number at the start (it is reasonable but it sounds like something is being covered up or avoided, which it is not)

304—what were the dates of the outliers, and why were they rejected?

310—are these the outliers (304) or is this something else?

Extended data map figures are excellent, very informative. If and when possible, it is great to have them compared at the same scale

Figures E 13, 14 (cross sections) need some kind of reference about vertical exaggeration, or the relationship between vertical scale and horizontal scale.

Figure E14, could the circle have degree marks (30, 45, 60, etc.)?

Supplementary Information

38—Is it sterile soil or (39) non-ceramic cultures; wording is confusing here.

163—are the walls 1m high and 10m wide (thick?); is this not more like a platform or causeway than a wall? Not clear. Has this been excavated to conclude that it has been infilled, or is this based on the mapping?

170—is the original depth measurement (2 or more meters) based on excavation or mapping?

186—was this site visited as well as mapped?

203—was this U-shaped rampart visited as well as mapped? Rampart as a word choice is a little unclear, maybe a closeup map would help.

245—this was obviously visited; can similar clarity be given for all the locations?

257—is there excavated evidence of a palisade or similar?

Supplementary table—this is very useful and important, could the coordinates be converted into DD.DDDD Lat/Long to make it even easier to examine satellite imagery? The table could also have 1, 2, 3, 4 tier designations. The * designation to indicate that these were unknown prior to Lidar-mapping is great.

Supplementary Table 5—could the number of settlements in each tier be counted?

Reference:

Souza, Jonas Gregorio, Denise Pahl Schaan, Mark Robinson, Antonia Damasceno Barbosa, Luiz EOC Aragão, Ben Hur Marimon Jr, Beatriz Schwantes Marimon, Izaias Brasil Silva, Salman Saeed Khan, and Francisco Ruji Nakahara. "Pre-Columbian Earth-Builders Settled along the Entire Southern Rim of the Amazon." *Nature Communications* 9, no. 1 (2018): 1125.

Referee #4 (Remarks to the Author):

Amazonian prehistory is being rewritten and these authors are part of this amazing transformation. I do believe that if this paper is revised the researchers should focus on being able to demonstrate a 4-tiered settlement organization rather than the overall size of the central ceremonial nodes. That is when placed in the context of the America's the size of those sites is very small. The important piece is how they fit into an organized, settlement/landscape pattern. This goes a long way toward demonstrating a much more complex socio-political organization than is commonly assumed for Amazonia. It would also be useful to learn more about the temporal affiliation of the ancient culture being discussed in this article. In terms of the supplementary material more information is needed about the way the scanner was used, the setup on the helicopter, resolution of the products, etc. Beach et al. have a very nice recent paper in PNAS that can be used as a guide for how to lay this out - Ancient Maya wetland fields revealed under tropical forest canopy from laser scanning and multiproxy evidence. In sum great paper.

Referee #5 (Remarks to the Author):

Using lidar to acquire precise information on large enclosed sites in SW Amazonia and mapping the connections between them.

The scale of the sites and the location substantially enhances our understanding of tropical societies and their ecological impact on the regional savanna. This is a novel study though it has antecedents further in to Amazonia.

The methodology of data acquisition is clear and appropriate and resolution limitations are reported. The lidar procedure is well established and valid. The quality of the data and the presentation is high.

No comment on statistics.

Please see sources in my attached document. No adverse criticism is intended of a high calibre, consequential paper. First I would note that the label low-density urbanism has now been extended to sites in Europe such as the European oppida (see Moore) and the Trypillia sites (Chapman and Gaydarska) which are of the same order of magnitude as the enclosures sites in the Llanos. I have recommended that a qualifier should be added just as we call megalopoli "industrial low-density urbanism" and places like Greater Angkor "agrarian-based low-density urbanism". As no qualifier is yet agreed for these smaller settlements - may I recommend that they be referred to in this paper as "a form of low-density urbanism".

I would avoid "mega-site" as it is used in many contexts world-wide eg for the 5-15 ha PPNB sites in the Levant.

I would recommend caution in the application of the term low-density urbanism to the c. 500 sq km around the big enclosed site - nor do I consider it necessary to make the point of the paper and use the term low-density urbanism just for the enclosed sites. The problem with the extended area is that it is in the order of magnitude of the Classic Maya sites and the great Sri Lankan Buddhist cities as well as earlier sizes of Greater Angkor. However, those sites have vast extended, landscape of numerous occupation mounds, shrines, water systems and agriculturally engineered landscapes. To add equivalent evidence to this paper eg the forest islands would demand substantial mapping and rigorous ground truthing of occupation dates which is an entire additional paper! I do not recommend that task for this paper to the authors but would therefore recommend that in this paper the point about the larger landscape be focussed on its interconnectedness. To claim that the c500 sq km are should be labelled low-density urbanism rather than a low-density urban settlement network seems to me to make a huge claim to equivalence with the urban organisational capacity of literate states. Note that if this same claim to extended low-density urbanism in these central localities is applicable to the eastern regions of the Llanos we would be seeing colossal low-density cities larger than Greater Angkor. That might be and it would be of profound importance - but to demonstrate even the 500 sq km extent would require as comprehensive a lidar and ground truthing procedure as has occurred for Caracol and Greater Angkor.

I suggest a more cautious, well-substantiated proposition in this paper.

In my attached document I have suggested some additional sources - please excuse my noting my own more recent papers. Roberts, Moore and Smith deserve noting. The issue that defining urbanism is now contentious should also be mentioned as it aids the case for using urban as a label in this paper.

The text and abstract are clear. I would recommend modifying the statements about what locations are being referred to as low density urbanism, avoid "megacities" and refer to the c 500 sq km area as a networked landscape or words to that effect.

Clarity and context: lucidity of abstract/summary, appropriateness of abstract, introduction and conclusions

Attached File:

Sources and additional information for the Llanos paper.

Information

Big enclosed sites (just for information. No need to use these)

Cornesti-Iacuri 17 sq km (15th-14th c BCE)

See Szentmiklosi et al 2011 Antiquity 85: 819-38

Sungbo's Eredo 1200 sq km (10th c CE) (not usually considered urban but surrounds a Yoruba town) See Darling 1998. Nyame Akuma 49: 55-61.

Iron Age Bagendon 170 ha. Chichester Walls 150 sq km (latter not called low-density urban)

References

Fletcher 1986 is superseded by Fletcher 1995

Fletcher R. 1995. The Limits of Settlement Growth. A theoretical outline. Cambridge: Cambridge University Press.

Fletcher 2009 is superseded by 2012

Fletcher R. 2012. Low-density, agrarian-based urbanism: scale, power, and ecology. In Michael E Smith (Eds.), The Comparative Archaeology of Complex Societies, (pp. 285-320). New York, USA: Cambridge University Press.

Fletcher 2019 needs Fletcher and White 2018 as that paper discusses the different magnitudes of tropical "megacities – low-density urbanism "

Fletcher R. and White K. 2018. Tropical environments and trajectories to low-density settlement forms. In Sanz N. (ed) Exploring Frameworks for Tropical Forest Conservation. Integrating natural and cultural diversity for sustainability: 92-115. A global perspective. UNESCO: Mexico City

Tom Moore to be added to Low-density urban commentary citing Gaydarska and Chapman

Moore T. 2017. Beyond Iron Age 'towns' Examining oppida as examples of low-density urbanism. Oxford Journal of Archaeology 36(3): 287-305.

On the issues of urban definitions

Fletcher 2020 and Smith 2020 are needed on the ambiguity of urban definitions.

Fletcher R. 2020. 'Urban Labels and Settlement Trajectories', Journal of Urban Archaeology, 1: 31-48. Smith M.E. 2020. Definitions and Comparisons in Urban Archaeology, Journal of Urban Archaeology 15-30

On transformations of tropical forest environments – see Patrick Roberts

Roberts P. 2019. Tropical Forests in Prehistory, History, and Modernity. Oxford: Oxford University Press.

Author Rebuttals to Initial Comments:

In this letter, we explain the revisions we have made in response to comments by the referees. We first address questions raised by most referees and then we respond to individual referees' comments following their order in the review. Each comment of the referee (in italics) is followed by our response (in normal text).

Use of the term megasite.

Referee #1:

This paper reports on new Lidar data for archaeological sites in the Llano de Mojos region of SW Amazonia. Amid a landscape already crowded with known archaeological sites, Lidar data were collected from 6 quadrants (2 large and 4 smaller) that revealed an additional 11 sites (from 15 to 26) and confirmed a 4-tier hierarchy of settlements dendritically connected via straight causeways. These somewhat modest gains are offset by recognition of the massive size of the largest sites (here termed megasites). Use of the term megasite is commendable and preferable over use of the term "city", which invites unresolvable polemics.

Similarly, Referee#3 states:

For megasites (outside Mojos) why is it that those sites are not just called cities? What are the main issues that the introduction alludes to about megacities and cities? How would Casarabe fit into that discussion?

And Referee#5:

I would avoid "mega-site" as it is used in many contexts world-wide eg for the 5-15 ha PPNB sites in the Levant'

Response: We are pleased that Referee#1 commend the use of the term megasite and agrees with us that terms like 'city' invites unresolvable polemics that we cannot answer with the current state of research in the region. Referee#5 critique does not apply since we made clear in the paper that our definition 'megasite' following Gaydarska (refs 24, 25) refers to sites larger than 100 ha.

Connections with the Andes and Pacific coast areas of South America.

Referee #1:

The monumental architecture--both U-shaped structures and conical mounds--indicates cultural linkages with the Andes and Pacific coast areas--a topic not pursued by the authors.

Similarly, Referee#3 states:

Two comparisons are made to the Andean archaeological record, which would be stronger if they were more explicit. The calculation of earth moved to create the Casarabe mounds in comparison to the Akapana is excellent, and could be pursued further: if assumptions are made about what large-scale architecture at Tiwanaku means, can those same assumptions be made about Casarabe? The proposed U-shaped constructions are a very interesting and important parallel to architecture in the highlands and on the coast; this evidence should be documented in greater detail.

...

U-shaped constructions appear to go back thousands of years in South America, is the Casarabe phenomenon part of this long-lived tradition?

...

The two potential supra-regional interpretations, one pointing toward the Andes and the other pointing towards the Amazon, should both be addressed. Mojos is sometimes in a strange position of being written out of histories of the Andes, because it is clearly not Andean, and also being written out of the Amazon, because it is a unique case, and at least for some people, not really Amazonian. Neither the Andean audience nor the Amazonian audience should be permitted to safely tuck this data away in a folder marked "anomalous" and therefore stop thinking about it. This research shows that Mojos is significant to both the Andes and the Amazon.

Response: At this stage of the research it is not possible to make any clear connection with either the Andes or the Pacific coast.

Comparison with Upper Xingu 'Garden Cities'.

Referee #1:

Data presentation and supplemental materials are clear and of pretty high quality. Referenced works are appropriate and fairly comprehensive but authors engage in what to my mind is needless hyperbole regarding SW Amazonia. Authors claim that their study represents "a new global case of tropical low-density urbanism and a first for Amazonia" (p. 2). Both Heckenberger and Erickson have presented site evidence from the Amazon that indicates the complexity, density, and greater size of Amazonian sites than previously imagined.

Referee#3 also asked for a clarification of this matter stating:

'Similarly, the paper uses the term urbanism, but does not make a detailed comparison to the Xingu example (which is cited in the abstract). Is the Casarabe case related to the Xingu case? If so, then how? Can specific comparisons in terms of size of constructions be made? How does this interpretation compare to the interpretations that have been made for the Xingu pattern? Souza et al. compare similar earthworks (or at least comparable earthworks)

across the entire Southern Amazon, this work should be mentioned and integrated into the discussion.'

Response: We agree that the work of Heckenberger and Erickson has indeed advanced our perception of the complexity, density and size of Amazonian sites, however, our fresh lidar data has shown that the Casarabe Tradition is qualitatively different from the Upper Xingu. To this end, we have added the following sentence in the concluding paragraph:

'We propose the Casarabe culture settlement system as a singular form of tropical agrarian low-density urbanism², the first known case for the entire tropical lowlands of South America^{29, 30}. The scale, monumentality and labour involved in the construction of the civic-ceremonial architecture, water management infrastructure, and spatial extent of settlement dispersal, compare favourably to Andean cultures and are to a scale far beyond the sophisticated, interconnected settlements of Southern Amazonia³¹, which lack monumental civic-ceremonial architecture. As such, the data contribute to the discussion of the global wealth of early urban diversity, and will help to redefine the categories used for past and present Amazonian societies.'

Definition of the settlement pattern tier hierarchy.

Referee #3:

The results presented here are significant to the interpretation of the archaeological record in eastern Bolivia, but also in the Andes and in the Amazon more generally. My main critique of the manuscript is not with the methods, the documentation, or the conclusions, but that the paper should do more, to make the case that these results have broader implications. The paper is based on a long and impressive record of research in the area. The conclusions of the paper draw on this record, in the form of site locations, excavation results, and radiocarbon dating. The manuscript should be published with revisions. Sharpening the focus of the discussion will give the paper more impact, and a larger audience. The paper reconstructs a four-tiered settlement hierarchy, and clearly lays out criteria for these four tiers (and a possible fifth).

Response: We welcome Referee #3 positive comments in terms of the methods, documentation, and conclusion. We are pleased to see that Referee#3 agrees with the clarity of our criteria to define the four-tier settlement system.

Why are the distinctions between the tiers made at those particular points; is the difference between 2nd and 3rd tier really significant? Why is a 2nd interpreted differently from a 3rd? Is the difference between 1 and 2 the same kind of difference as the difference between 3 and 4?

112-122—would it be possible to know more about how these classes were selected, why these are significant “break” points?

Response: We thought carefully about the definition of the settlement tiers.

We need to bear in mind that the sample size of sites where we have detailed digital terrain models produced by lidar including height, core platform area and area enclosed by the polygonal enclosures is ten, which hardly represent an adequate sample size for sound statistical analysis. Bearing this in mind, if we carry out a k-mean cluster taking into account core area and polygonal enclosure area from the 10 sites we have data from.

	Inner	Poly
Tier 1	12-22.3	147.2-314.8
Tier 2	2.6-6.52	15.9-47.3
Tier 3	0.5-1.2	2.3-5.1

(Inner=core area; Poly=Polygonal enclosure)

The statistic comes up with three significant groups, which are previously defined Tier 1, Tier 2, and 3-4 combined. We consider it important to keep the distinction between Tier3 and Tier 4 since Tier 4 sites do not exhibit a polygonal enclosure nor do they have any type of mounded architecture on top of the core area.

When we run the k-mean cluster analysis for all the sites where we have at least core area – polygonal area scores 0 when is not present-, which total 19 sites, then the k-clusters come up with more than 8 clusters (see below), so we discarded this classification.

Therefore, with the data at hand, we decided to define a settlement hierarchy based on quantitative parameters (core area, polygonal enclosure area, height, number of polygonal enclosures, number of straight causeways) and qualitative ones (diversity of civic-ceremonial architecture) (Table S5). Future lidar data from the region will certainly corroborate or refined this settlement pattern tier system.

REFEREE #1

*Lidar data were collected from 6 quadrants (2 large and 4 smaller) that revealed an additional 11 sites (from 15 to 26) and **confirmed** a 4-tier hierarchy of settlements dendritically connected via straight causeways. (emphasis added).*

Response: We would like to make the clarification that our novel data has not confirmed, but defined a new 4-tier hierarchy of settlements. Previous coarse remote sensing data only allowed for a two-tier hierarchy: mounds and forest islands (Lombardo and Prümers, 2010). Therefore, the new classification is a major step change.

This work builds on those studies and indicates that SW Amazonia was not a sparsely settled backwater but a densely settled and hierarchically organized part of the larger Amazonian world.

Response: Agreed.

REFEREE #3

In his concluding comments, Referee #3 states *'It is a great credit to the research and the manuscript that it leads to these kinds of discussions. **Because of space limitations it is might not possible to address all of these larger points. I would advocate for choosing at least one of the implications and pursuing in more detail, and then leaving the others as open questions that can be returned to in future research.**'* (emphasis added).

We welcome Referee#3 appreciation of space limits. We are not concentrating on one point, and we are addressing all his/her suggestions below:

What would make this argument stronger would be to include a map or at least counts of how many sites are included in each category, and how they are arranged on the ground. For example, at Cotoca, one 1st tier site is said to be accompanied by 3 2nd tier sites, 2 3rd tier sites, and (presumably) 13 4th tier sites. More information about the arrangement of these sites and their relationships with one another are needed if the settlement hierarchy is going to support a strong interpretation.

Response: We believe that with the data at hand, we have done an exhaustive description of the sites and their connectedness from the areas where we have the best regional data including the Cotoca cluster (Figures 2-3, Extended Data Figures 1, 8, 10-11) and the El Cerrito cluster (Extended Data Figure 3, 9).

In the past, 4-tier settlement hierarchies have been held to stand for 4-tiered political organizations (like states), is that what the authors are proposing? This is not clear as written, and the paper would be strengthened if it contained a stronger statement about what the pattern represents. Does the spatial hierarchy represent a social hierarchy?

Response: We believe that it is too early in the state of research in the region to make claims about Archaic states. It is beyond the scope of this paper to produce such speculations at the moment.

A skeptical audience might imagine that what appears to be a U-shape on the map is only a mound that has been dug into, or eroded, for example. A little more detail would disarm such criticism.

Response: We have added a new illustration that makes the U-shaped layout of those buildings clearly visible.

The authors are unequivocal that this paper settles the question of whether western Amazonia was sparsely settled before 1492, and I agree. I think that this point would hit home for the non-specialist reader with a brief explanation of the potential of the LIDAR methodology in other forested locations.

Response: Due to space limitations, we cannot elaborate on the potential of lidar methodology in other regions of the world. However, in our introductory sentence ‘Lidar has recently revealed ancient low-density agrarian urban societies beneath the tropical forests of Asia, Africa, and Central America¹⁻⁶.’, we made the point succinctly.

A single sentence about how the LIDAR plots were chosen, and (presumably) the earthworks that were mapped were not visible on other satellite or aerial photography, would then lead the reader to the question of how many more earthworks are present not only in eastern Bolivia (ie in the maps presented here), but in the western Amazon more generally (and perhaps across the entire Amazon). This region is comparatively well studied, after all, and the current paper makes it clear that many earthworks had gone undetected. (emphasis added).

Response: We do not understand this criticism (in bold above), we clearly state in the paper that we selected areas with known concentrations of major settlements. As written in the paper: ‘To remedy this 62 situation we conducted airborne laser mapping for six areas (10-85 km²) **with known concentrations of major settlements**, totalling 204 km² (Fig. 1).’ (emphasis added).

Notes from the manuscript, figures, and supplementary information (by line number)

37—not so much a “global case” as a case that fits into some larger idea?

Response: Agreed. We have rephrased the sentence and it now reads: “Our results indicate that the Casarabe culture settlement pattern represents a new case of tropical low-density urbanism of a kind previously unknown in Amazonia .”

46—is this meant as a difference from other areas within the LM?

Response: In this passage, we are referring to the Llanos de Mojos as a whole region and the sentence is quite clear in saying, that within this larger area we have "diversity in socio-political organisation, water control systems, and economic bases".

79-80—they may have been known, but not as megasites; this undersells the importance of the research.

Response: We think that our text is clear enough in this regard when we state ‘Both megasites were already known, but their massive size and architectural elaboration only became apparent through the lidar survey.’

84—is there any evidence of palisades in the area, or other corroborating evidence for an interpretation of these earthworks as defensive structures?

Response: No, at the moment, there is no evidence of palisades or other defensive structures and that's why we did not include any.

86—The presentation of the U-shaped buildings needs more detail because of the interpretive weight that these constructions carry (esp. among Andeanists)

Response: We have produced a new figure.

107—comparison to Akapana is excellent, are there any other comparisons (Xingu? Maya? SE Asia) that could be briefly added here to supplement the comparative perspective (which the abstract seems to promise).

Response: Comparison with the Uppper Xingu, the other region in Amazonia has been addressed. It is beyond the scope of this paper to make detailed comparisons with the Maya or SE Asia.

138—if there are 3 secondary and 2 tertiary, does this have any implications for the interpretation of the four tier hierarchy?

Response: Unfortunately, our sample is too small to tackle this kind of nuances.

151—this is an excellent point and the paper could even go farther in this direction with more information about how many earthworks were found in the given area, and then discuss how much more could be found not just in eastern Bolivia but in the western Amazon more generally.

Response: We agreed, but 'going further', which we guess the referee refers to modellings like co-author Iriarte and his team has done for the southern rim of the Amazon (Souza et al. 2018, Nature Communications) is beyond the scope of this paper.

Figure 1—strongly makes the point that the LIDAR coverage only includes a fraction of the area; this could be emphasized more strongly in the caption or somewhere in the text.

Response: We believe this is self-explanatory and clear to see to the reader.

Figure 2—this figure is excellent; perhaps add a legend for the color scale/elevation.

Response: We have added a colour scale for elevation to the figure legend.

Recognizing the problem of space, is it possible to indicate which mounds or earthworks were unknown? Perhaps in the text or caption? (this is clear in the supplemental information on the list, but maybe it can be shown here)

Response: As Referee#3 states this information is readily available in the Supplemental Information. It is also now shown in Figure 1.

Figure 3—Good image, this multiplies the impact of Figure 2 because it shows that it is not a unique case.

Response: Agreed.

298—state the number at the start (it is reasonable but it sounds like something is being covered up or avoided, which it is not)

Response: We have amended the sentence, which now reads:

‘Excavations at monumental sites (locally called 'Lomas'), Salvatierra and Mendoza have revealed complex stratigraphies that bracket the chronology of the Casarabe culture from AD 500 to 1400^{10-12, 40, 42}. The 94 radiocarbon dates from Salvatierra and Mendoza¹², as well as 50 additional radiocarbon dates previously published for other Casarabe culture sites^{40, 42, 44}, date the construction and use of these sites to this period (Supplementary Tables 2-4, Extended Data Figs. 17-19).’

304—what were the dates of the outliers, and why were they rejected?

Response: Both rejected dates predate the start of settlement by over a thousand years and come from contexts correctly dated by other C14 dates. One of the outlier dates comes from the Mendoza site and is discussed in detail by Prümers (2015: 82). While all other dates from Mendoza fall between AD 600-1400 this date (Bln 5328) was 2800±29 BP (cal BC 1000-820; 2 σ). The other outlier is from the Salvatierra site (MAMS 48305; 2995±25 BP; cal AD 1279-1023, 2 σ) and came from a context that, judging by the archaeological evidence, belongs to phase 3, dated between cal AD 750-1000.

This section now reads:

‘The dates prior to this period are mostly outliers but, in some cases, also come from samples taken from the sterile soil beneath the earliest platform buildings. They date natural events or perhaps the ephemeral presence of non-ceramic cultures prior to the construction of the Casarabe culture sites.’

310—are these the outliers (304) or is this something else?

Response: No, these are two dates from the Chocolatelito site, that are unusually “early”.

Extended data map figures are excellent, very informative. If and when possible, it is great to have them compared at the same scale

Response: The sites documented differ considerably in size. We had to adapt their scale to make them fit into the given space of a page. The scale of the maps has been selected so that the reader can better appreciate the details of the architecture of the sites.

Figures E 13, 14 (cross sections) need some kind of reference about vertical exaggeration, or the relationship between vertical scale and horizontal scale.

Response: Both scales are indicated in the figures so their relationship can be easily deduced.

Figure E14, could the circle have degree marks (30, 45, 60, etc.)?

Response: We added the degree marks to Figure ED 14.

Supplementary Information

38—Is it sterile soil or (39) non-ceramic cultures; wording is confusing here.

Response: We refer to sterile soil and it is clear in our text that reads:

‘The dates prior to this period are mostly outliers but, in some cases, also come from samples taken from the sterile soil beneath the earliest platform buildings. They date natural events or perhaps the ephemeral presence of non-ceramic cultures prior to the construction of the Casarabe culture sites.’

163—are the walls 1m high and 10m wide (thick?); is this not more like a platform or causeway than a wall? Not clear. Has this been excavated to conclude that it has been infilled, or is this based on the mapping?

Response: From the context of the entire complex, it can be concluded that these are ruined defensive structures. The walls (of adobe or tapia) were probably several metres high and the cone of debris caused by their collapse is correspondingly wide. Since none of these structures have been excavated, the interpretation may of course change with new findings.

170—is the original depth measurement (2 or more meters) based on excavation or mapping?

Response: No archaeological excavations have yet been carried out at the Cotoca site. The estimate is based on data from the Salvatierra site, where trench excavations of eroded ditches prove their original depth of up to 2m.

186—was this site visited as well as mapped?

Response: No, the Landivar megasite has just been mapped with lidar and detailed mapping and visits of the site have not yet been carried out. The detailed digital terrain model produced by lidar will inform future targeted ground-based research.

203—was this U-shaped rampart visited as well as mapped?

Response: Yes, it was visited but only cursorily because this part of the site is swampy almost all year round.

203— Rampart as a word choice is a little unclear, maybe a closeup map would help.

Response: Agreed. We changed the sentence: “Another remarkable feature is the area bounded by a U-shaped embankment (12) to the west of the base area platform.”

245—this was obviously visited; can similar clarity be given for all the locations?

Response: All sites in our “List of Casarabe culture sites” (SI Table 1) were visited with the exception of those marked as “unknown prior to Lidar-mapping”.

257—is there excavated evidence of a palisade or similar?

Response: No, there is no evidence of palisades although at the Salvatierra site we made several excavation cuts in the polygonal rampart to clarify whether palisades had been erected there.

Supplementary table—this is very useful and important, could the coordinates be converted into DD.DDDD Lat/Long to make it even easier to examine satellite imagery?

Response: The archaeological sites of the Llanos de Mojos are usually not visible on satellite images because they lie under dense vegetation. For this reason, the use of LIDAR made sense. If a reader wants to view the sites in Google Earth, he/she can easily use any of the online geographical coordinate converters (e.g. <http://rcn.montana.edu/Resources/Converter.aspx>).

*The table could also have 1, 2, 3, 4 tier designations. The * designation to indicate that these were unknown prior to Lidar-mapping is great.*

Response: Done.

Supplementary Table 5—could the number of settlements in each tier be counted?

Response: Due to the patchy nature of our lidar data this cannot be done at the moment.

Reference:

Souza, Jonas Gregorio, Denise Pahl Schaan, Mark Robinson, Antonia Damasceno Barbosa, Luiz EOC Aragão, Ben Hur Marimon Jr, Beatriz Schwantes Marimon, Izaias Brasil Silva, Salman Saeed Khan, and Francisco Ruji Nakahara. “Pre-Columbian Earth-Builders Settled along the Entire Southern Rim of the Amazon.” Nature Communications 9, no. 1 (2018): 1125.

This reference lacks the senior author of the paper Prof. Iriarte.

Referee #4 (Remarks to the Author):

*Amazonian prehistory is being rewritten and these authors are part of this amazing transformation. **I do believe that if this paper is revised the researchers should focus on being able to demonstrate a 4-tiered settlement organization rather than the overall size of the central ceremonial nodes.** That is when placed in the context of the America’s the size of those sites is very small. The important piece is how they fit into an organized, settlement/landscape pattern. This goes a long way toward demonstrating a much more complex socio-political organization than is commonly assumed for Amazonia.*

Response: We regret to disagree with Referee#4 criticism (in bold). The discovery of the megasites is the major finding of the paper. We do not believe that the size of the ‘central ceremonial nodes’ is very small. The megasites are two orders of magnitude larger than the ‘Garden Cities’ of the Upper Xingu and 10 times larger than Tiwanaku’s Akapana. Similarly, we disagree with the statement of Referee#4 that “*when placed in the context of the America’s the size of those sites is very small*”. Of course, there are much larger sites in the Americas than the ones we discover with lidar in the Llanos de Mojos. However, the Casarabe culture megasite compare similarly to other low-density urban cultures in tropical America. For example, for the Maya area, Smith (“How can Archaeologists identify early cities?” In: Eurasia at the Dawn of History: Urbanization & Social Change. Manual Fernández-Götz & Dirk Krausse, eds. Cambridge Univ. Press.2016:162) indicates areas of 1 ha for villages, 15 ha for towns, and 210 ha for city-state capitals. The core area of the Cotoca site, with the civic-ceremonial monumental architecture alone, covers 15 ha and would, according to the ranking of Smith, classify as a “town”. Cotoca’s total area is 147 ha, a value very close to that given for the city-state capitals.

It would also be useful to learn more about the temporal affiliation of the ancient culture being discussed in this article.

Response: We believe we clearly bracketed temporally the Casarabe Tradition and have dedicated a whole supplemental information section to it. This is clear for the reader when we state: ‘The Casarabe culture developed here between ~AD 500 and 1400, over an area of 4,500 km² (see Chronology in supplementary text, Supplementary Tables 2-4, Extended Data Figs. 16-18).

In terms of the supplementary material more information is needed about the way the scanner was used, the setup on the helicopter, resolution of the products, etc. Beach et al. have a very nice recent paper in PNAS that can be used as a guide for how to lay this out - Ancient Maya wetland fields revealed under tropical forest canopy from laser scanning and multiproxy evidence. In sum great paper.

Response: We agreed. We have added more information about the equipment, the PPR used, airspeed, and transect overlap. It now reads: ‘The sensor used was a Riegl VUX-1 scanner, with a Trimble APX-15 UAV GNSS, attached to a Eurocopter AS350 helicopter using a custom mount. Laser Pulse Repetition Rate (PRR) was 200 kHz. Flight altitude was 200 m above ground level, airspeed was 45 knots. Missions were flown in 200 m parallel strips, with 50% overlap.’

Referee #5

Using lidar to acquire precise information on large enclosed sites in SW Amazonia and mapping the connections between them.

The scale of the sites and the location substantially enhances our understanding of tropical societies and their ecological impact on the regional savanna. This is a novel study though it has antecedents further in to Amazonia.

The methodology of data acquisition is clear and appropriate and resolution limitations are reported. The lidar procedure is well established and valid. The quality of the data and the presentation is high.

No comment on statistics.

Response: We are pleased with the complimentary comments of referee #5.

Please see sources in my attached document. No adverse criticism is intended of a high calibre, consequential paper. First I would note that the label low-density urbanism has now been extended to sites in Europe such as the European oppida (see Moore) and the Trypillia sites (Chapman and Gaydarska) which are of the same order of magnitude as the enclosures sites in the Llanos.

Response: Unfortunately, due to space limits, we cannot include all the references on early urban cultures across the world.

I have recommended that a qualifier should be added just as we call megalopoli "industrial low-density urbanism" and places like Greater Angkor "agrarian-based low-density urbanism". As no qualifier is yet agreed for these smaller settlements - may I recommend that they be referred to in this paper as "a form of low-density urbanism".

Response: We agreed with the point brought up by referee #5, the concluding sentence now reads: 'We propose the Casarabe culture settlement system as a singular form of tropical agrarian low-density urbanism², the first known case for the entire tropical lowlands of South America^{29, 30}'

I would recommend caution in the application of the term low-density urbanism to the c. 500 sq km around the big enclosed site - nor do I consider it necessary to make the point of the paper and use the term low-density urbanism just for the enclosed sites. The problem with the extended area is that it is in the order of magnitude of the Classic Maya sites and the great Sri Lankan Buddhist cities as well as earlier sites of Greater Angkor. However, those sites have vast extended, landscape of numerous occupation mounds, shrines, water systems and agriculturally engineered landscapes. To add equivalent evidence to this paper eg the forest islands would demand substantial mapping and rigorous ground truthing of occupation dates which is an entire additional paper! I do not recommend that task for this paper to the authors but would therefore recommend that in this paper the point about the larger landscape be focussed on its interconnectedness. To claim that the c500 sq km are should be labelled low-density urbanism rather than a low-density urban settlement network seems to me to make a huge claim to equivalence with the urban organisational capacity of literate states. Note that if this same claim to extended low-density urbanism in these central localities is applicable to the eastern regions of the Llanos we would be seeing colossal low-density cities larger than Greater Angkor. That might be and it would be of profound importance - but to demonstrate even the 500 sq km extent would require as comprehensive a lidar and ground truthing procedure as has occurred for Caracol and Greater Angkor.

Response: We agreed with the reviewer, but yet again, given the nature of the data and the early stage of research in the region, this cannot be addressed and is beyond the scope of this paper.

In my attached document I have suggested some additional sources - please excuse my noting my own more recent papers. Roberts, Moore and Smith deserve noting. The issue that defining urbanism is now contentious should also be mentioned as it aids the case for using urban as a label in this paper.

Response: We thank the reviewer for the update on the references, all of which we are aware of, and include in the upcoming discipline-specific papers that allow us more space for citations.

The text and abstract are clear. I would recommend modifying the statements about what locations are being referred to as low density urbanism, avoid "megasites" and refer to the c 500 sq km area as a networked landscape or words to that effect.

Response: We have addressed these issues above.

We hope that we have satisfactorily responded to the concerns of the reviewers and that our manuscript is now acceptable for publication in Nature.

Sincerely,

For the authors,
Heiko Prümers

Reviewer Reports on the First Revision:

Referees' comments:

Referee #1 (Remarks to the Author):

I have re-read this manuscript as well as the extensive responses by the authors to comments in the first round of reviews. Given the restrictions for article length imposed by Nature, I am satisfied that the current ms. is sound and makes an excellent and new contribution to Amazonian archaeology. I recommend its publication.

Referee #3 (Remarks to the Author):

The authors have addressed all the questions that I raised in my earlier review, and the paper should certainly be published in Nature as revised. It presents new information that deepens our understanding of monumental earthworks and settlement in the Llanos de Mojos, the Amazon, and beyond.

The authors have responded in a careful, thorough, and professional manner to the reviewers' comments. As the authors indicate, it is not possible to address all of the many issues concerning interaction with other places in the Andes and the Amazon, the definitions of urban or megasite, and the connection between settlement hierarchies and social organization. The article is significant because it promises to encourage further research around these questions. It is difficult to imagine a stronger reason in favor of its publication.

I apologize for misrepresenting the recommended article and omitting the name of the senior author; it was wholly unintentional.

All the authors are to be congratulated on their excellent research and for writing a significant paper.

Referee #5 (Remarks to the Author):

My comments are brief. Within the paper constraints the authors have clearly presented their results. The discussion of these sites will be affected by terminological inconsistencies which should be removed as it is simple to do. Megasite is not an appropriate term and is only being used for its Classical word cachet. Gaydarska and Chapman now use the term First Cities for the Trypillia sites and refer to them as urban. Recently another paper on the Trypillia sites was criticised for using the term megasite. As I noted previously priority for that term goes to the PPNA sites which are quite small and are not at all the equivalents of these Amazonia sites.

Megasite is unnecessary. Giant is nicely expressive. Large sites would suffice.

I suggest that you also delete Ref 1 to Fletcher 1986 as it is not pertinent to the low density urban discussion.

Referee #7 (Remarks to the Author):

Thank you for the opportunity to read this interesting study. In accordance with Nature's review policy, I have aggregated my comments into the following sections:

A. Summary of Key Results

This is a well-written study that uses derivative topographic products from an airborne LiDAR survey to examine areas with known monumental architecture in SW Amazonia to detail the morphology of those sites and reveal associated settlement patterns which were previously unknown.

B. Originality and Significance

Using LiDAR to study human impacts on landscapes and cultural landscape features in forested regions of the world has indeed become very common over the past ~10 years or so (also in North America and Europe, which are not mentioned by the authors). Therefore, the methods described in this paper are not novel on a broader scale, but appear to be the first application in this region of the world. Additionally, the information derived from the LiDAR survey allowed for subsequent digitization of landscape features and further analyses of settlement density and site characterization (the four-tiered structure). In terms of studies which use LiDAR, the study's contribution to the literature comes broadly as a case study depicting unique types of land use features in a specific part of the world, with further significance since it also advances the understanding of settlement patterns in this part of the world and their relationship to other similar settlement patterns elsewhere.

C. Data and Methodology

The LiDAR Methods section for this paper is missing several standard pieces of information, and in my opinion this information is needed prior to publication. The section should describe how the point cloud was classified, filtered (i.e., ground-classified points only?), and the interpolation process/parameters that were used to generate the digital elevation models (DEMs) used in analysis, as well as any effects that vegetation in the region could have had in the collection of the data and classification of the point cloud itself. Additionally, this section should note the point density of the main and filtered point cloud used in DEM generation, as well as the pixel size of the resulting DEM. In many regions, low vegetation can be mistaken for the ground surface in classification algorithms which can make it challenging to discern detail; further, if vegetation prevents the pulses from reaching the actual ground surface, then there can be issues with interpolation and subsequent feature identification if the point density of ground-classified points is not high enough. Please also be specific about the visualization techniques that you used in your analysis (e.g. hillshade, slope, sky view factor, etc.) as there are quite a variety and they offer various benefits to visualizing different types of cultural landscape features.

In terms of how the authors selected LiDAR survey locations, it would be helpful to have a bit more detail if possible. I see that the text says the reasoning for selection as being "known concentrations of major settlements" - it would be helpful to know if those were the same areas called out in figures S5 and S7 that were already well known/mapped, so they could be compared to the LiDAR results in at least one case, for example. In Figure 1, I looked at area A which has 3 sites; 53, 54, 55. I assumed these referred to sites listed in Supplemental Table 1. The names all have the word "Lidar" in front of

them with an asterisk which indicates they were unknown prior to your mapping. Following that logic, there would have been no sites in Area A and B that were known prior to the LiDAR survey. Can you please clarify or provide more detail about the survey process and site selection?

D. Appropriate Use of Statistics/Uncertainty

Regarding uncertainty, I must ask the authors if these features been validated in the field to some degree? I recommend adding a statement regarding this in the text somewhere. I see that Extended Figure 7A and 9D are compared, but it is a challenge to try to discern any differences or similarities. A more direct comparison between field-based data and LiDAR-based data somewhere, especially some acknowledgement of validation, is recommended.

E. Conclusions: robustness, validity, reliability

I see no issues with the validity of the conclusions reached in this article. Nuances in the LiDAR point cloud and topographic derivatives could have some impact on the numbers presented in the volumetric or area calculations, but I wouldn't expect it to change the nature of their conclusions in any significant way unless there were serious issues with the way the point cloud was classified or filtered.

F. Suggested Improvements

In addition to my comments in the other sections, I also have the following comments:

Line 58: The text notes that "only a handful of isolated sites" provide examples of mounded architecture (citing Figs S5 and S7), but Figure 1 shows ~180 or so sites scattered throughout the study region with black triangles noted as being 'monumental mounded architecture' in the caption. Just to clarify, do you mean that the only sites of these ~180 that have been mapped in any detail are the sites shown in Figures S5 and S7? If so, please clarify that in the text. Can you also please note how Salvatierra (S5) was mapped in 2004-2006?

Line 64: The text says "...the two megasites" but up to this point it is unclear what sites you are referring to and whether they were previously documented or if you found new ones. I recommend the following wording, "Lidar documented in detail two known megasites (Cotoca and Landivar), as well as 24 smaller sites..." I have found myself using Figure 1 and Supplemental Table 1 to follow along with the geography of your study region, so think it would be helpful to have in parentheses the names for comparison. Also, are these the only megasites in this region, or are there others? You might consider adjusting Figure 1 to show which sites you confirmed using LiDAR, which were new sites, and also the location of megasites in the region.

Figure 2 and other Extended Data LiDAR figures: These figures (or at least a few) could be improved to label types of observed cultural features. At the moment, we see the morphology of various features in Figure 2, but to someone not familiar with the landscape in this region, it is not obvious what they are. Features are labeled in Extended Data Figure 1 with boxes and numbers, but the numbers are not described in the caption directly, so it's not really made clear what the objects are. There are several figures in the Extended Data where features are numbered but not described in the caption or directly cited in the text. I do see some areas in the text where features are described and the figure is cited, but since there will likely be readers who are not familiar with this type of

architecture or features, I would recommend ensuring all of your Extended Data captions make note of what the numbered/lettered cultural features are so it is clear to the reader.

G. References

Credit to previous work is sufficiently cited in my opinion.

H. Clarity/Context

Overall, the study is well-written and the structure works well. There do seem to be some areas where I found the text to read somewhat awkwardly. For example, the sentence in the abstract “Here we present lidar data from the Casarabe culture...” suggests that the LiDAR data itself comes from that culture. I would suggest changing the sentence to reflect that LiDAR data was collected in a region of the world where that culture lived, as an example of one grammatical change. The title wording “Lidar evidence of megasites...” also does not seem correct, and I’d suggest considering something such as “Lidar reveals evidence of megasites...”

Author Rebuttals to First Revision:

In this letter, we explain the revisions we have made in response to comments by the referees #5 and #7. Each referee's comment (in italics) is followed by our response (in normal text).

We thank Reviewers #1 and 3 for their highly positive comments in relation to our revised manuscript. We are delighted they found it to be of excellent quality, clear, and comprehensive. We are encouraged that they consider our manuscript a new and significant contribution to Amazonian archaeology and beyond.

Referee #5 / Use of the term megasite:

Megasite is not an appropriate term and is only being used for its Classical word cachet. Gaydarska and Chapman now use the term First Cities for the Trypillia sites and refer to them as urban. Recently another paper on the Trypillia sites was criticised for using the term megasite. As I noted previously priority for that term goes to the PPNA sites which are quite small and are not at all the equivalents of these Amazonia sites.

Megasite is unnecessary. Giant is nicely expressive. Large sites would suffice.

Response: Agreed. Throughout the text, we have changed “megasite” to “large settlement site”.

We also modified the text lines 75-80, so it states now:

“Settlements that, at over 100 ha in size, exceed most other settlements of the same culture many times over, are a very early and worldwide phenomenon^{2, 24, 25}. A formally agreed term for these sites is still missing²⁶, so in this paper, we use the descriptive term “large settlement site” when referring to the two most important sites in the region: Cotoca (147 ha; Figs. 2, S1) and Landívar (314 ha; Figs. 3, S2). These two large settlement sites were already known, but their massive size and architectural elaboration only became apparent through the lidar survey.”

Referee #5:

I suggest that you also delete Ref 1 to Fletcher 1986 as it is not pertinent to the low density urban discussion.

Response: Agreed. We have deleted Ref 1 and we have split the references of the first sentence:

“Lidar has recently revealed ancient low-density agrarian urban societies¹⁻² beneath the tropical forests of Asia, Africa, and Central America³⁻⁶.”

1. Fletcher, R. Low-Density, Agrarian-Based Urbanism: A Comparative View. *Insights* 2, 2–19 (Durham University, 2009).

2. Lucero, L. J., Fletcher, R., Coningham, R. From ‘collapse’ to urban diaspora: the transformation of low-density, dispersed agrarian urbanism. *Antiquity* **89**, 1139-1154 (2015).³ Evans, D.H., Fletcher, R.J., Pottier, C., Chevance, J.-B., Soutif, D., Tan, B.S., Im, S., Ea, D., Tin, T., Kim, S. Uncovering archaeological landscapes at Angkor using lidar. *Proceedings of the National Academy of Sciences* **110**, 12595-12600 (2013).
4. Davis, S. D., Douglass, K. Aerial and Spaceborne Remote Sensing in African Archaeology: A Review of Current Research and Potential Future Avenues. *African Archaeological Review* **37**, 9-24 (2020).
5. Canuto, M.A., Estrada-Belli, F., Garrison, T.G., Houston, S.D., Acuña, M.J., Kováč, M., Marken, D., Nondédéo, P., Auld-Thomas, L., Castanet, C. Ancient lowland Maya complexity as revealed by airborne laser scanning of northern Guatemala. *Science* **361** (2018).
6. Chase, A.F., Chase, D.Z., Fisher, C.T., Leisz, S.J., Weishampel, J.F. Geospatial revolution and remote sensing LiDAR in Mesoamerican archaeology. *Proceedings of the National Academy of Sciences* **109**, 12916-12921 (2012).

Referee #7 / Lidar Methods:

The LiDAR Methods section for this paper is missing several standard pieces of information, and in my opinion this information is needed prior to publication. The section should describe how the point cloud was classified, filtered (i.e., ground-classified points only?), and the interpolation process/parameters that were used to generate the digital elevation models (DEMs) used in analysis, as well as any effects that vegetation in the region could have had in the collection of the data and classification of the point cloud itself. Additionally, this section should note the point density of the main and filtered point cloud used in DEM generation, as well as the pixel size of the resulting DEM. In many regions, low vegetation can be mistaken for the ground surface in classification algorithms which can make it challenging to discern detail; further, if vegetation prevents the pulses from reaching the actual ground surface, then there can be issues with interpolation and subsequent feature identification if the point density of ground-classified points is not high enough. Please also be specific about the visualisation techniques that you used in your analysis (e.g. hillshade, slope, sky view factor, etc.) as there are quite a variety and they offer various benefits to visualising different types of cultural landscape features.

Response: Agreed. We have added the following information to the methods section (Line 291 ff.):

“Raw point cloud density varied between 13 to 20 million pts./km², but generally was about 18 million pts./km². The filtering was done automatically taking into account from the outset only last pulses and points with only one reflection. The macros created to pre-classify the point cloud were tested using tiles that best reflected the nature of the terrain. Results were then reviewed and modified until an optimal result with only minor residual errors was achieved. At the end of this process DSM las-files were generated that had a mean point spacing of 0,3m. From these DEMs with 50 cm/pixel were generated using the “Natural Neighbors” Method (ArcMap). We used visualisation techniques provided by ArcMap (hillshade, slope) and the Relief Visualization Toolbox (RVT_2.2.1.) developed by the

Research Center of the Slovenian Academy of Sciences and Arts³⁷⁻³⁹. Display options were chosen in such a way that they led to an optimal visibility of the archaeological remains.”

With regard to the reviewers comment “*Please also be specific about the visualisation techniques that you used in your analysis (e.g. hillshade, slope, sky view factor, etc.) as there are quite a variety and they offer various benefits to visualising different types of cultural landscape features*” we would like to point out that we already state the use of:

“...visualisation techniques provided by ArcGIS and the Relief Visualization Toolbox (RVT_2.2.1.) developed by the Research Center of the Slovenian Academy of Sciences and Arts³⁷⁻³⁹.”

The RVT_2.2.1. provides a whole range of combinations already optimised for visualising archaeological structures. We have tested all possibilities for an optimal result in each individual case and maps differ accordingly. Sometimes differences are mere nuances in % of display transparency.

We hope that reviewer #7 agrees with us that the crucial point is that ‘The display options were chosen in such a way that they led to an optimal visibility of the archaeological remains on the plans’. This clarifying statement was added to the Methods section (see above).

Referee #7/ Selection of Lidar survey locations:

In terms of how the authors selected LiDAR survey locations, it would be helpful to have a bit more detail if possible. I see that the text says the reasoning for selection as being “known concentrations of major settlements” - it would be helpful to know if those were the same areas called out in figures S5 and S7 that were already well known/mapped, so they could be compared to the LiDAR results in at least one case, for example. In Figure 1, I looked at area A which has 3 sites; 53, 54, 55. I assumed these referred to sites listed in Supplemental Table 1. The names all have the word “Lidar” in front of them with an asterisk which indicates they were unknown prior to your mapping. Following that logic, there would have been no sites in Area A and B that were known prior to the LiDAR survey. Can you please clarify or provide more detail about the survey process and site selection?

Response: Site #55 is listed in Table 1 as “La Punta”, without an asterisk. It is the site where one of the authors (C.J.B.) has conducted fieldwork over the last years and the main reason to include this area for Lidar mapping.

In general, it should be noted that the selection of the areas was based on twenty years of work in the region. In addition to strictly scientific criteria, we chose locations with known major sites and tightly packed with Casarabe culture archaeological sites near Trinidad airport, which serves as the helicopter’s base. Far-flung places from Trinidad would have inflated the lidar operation’s expenditures unjustifiably:

Referee #7: Appropriate Use of Statistics/Uncertainty:

“Regarding uncertainty, I must ask the authors if these features been validated in the field to some degree? I recommend adding a statement regarding this in the text somewhere. I see that Extended Figure 7A and 9D are compared, but it is a challenge to try to discern any differences or similarities. A more direct comparison between field-based data and LiDAR-based data somewhere, especially some acknowledgement of validation, is recommended.”

Response: Agreed. We have added figure 5B to directly compare the Lidar-Data with those from terrestrial mapping (done with a total station Leica TPS 800).

Figure caption was modified to:

“Extended Data Fig. 5 | Maps of the Salvatierra site (#108). A, DEM calculated from 20,000 points measured with a total-station (Leica TPS 800) during seven months of surveying (campaigns 2004–2006). B, High-resolution lidar image (DSM; 50cm/pixel).”

Referee #7: Suggested Improvements

“Line 58: The text notes that “only a handful of isolated sites” provide examples of mounded architecture (citing Figs S5 and S7), but Figure 1 shows ~180 or so sites scattered throughout the study region with black triangles noted as being ‘monumental mounded architecture’ in the caption. Just to clarify, do you mean that the only sites of these ~180 that have been mapped in any detail are the sites shown in Figures S5 and S7? If so, please clarify that in the text. Can you also please note how Salvatierra (S5) was mapped in 2004–2006?”

Response: Yes, only the sites shown by the maps included in this paper have been mapped in any detail. The caption of Fig. 1 has been changed to:

“Fig. 1. Map of the southeastern Llanos de Mojos. Lidar coverage is marked by the gray areas (A–F). Black triangles represent settlement sites of the Casarabe culture with platform mound architecture. The topographic layer is based on TanDEM-X DEM 12m data.”

As stated above, Fig. 5 was changed, and information regarding the surveying technique used in 2004–2006 at the Salvatierra site was added to the figure capture.

“Line 64: The text says “...the two megasites” but up to this point it is unclear what sites you are referring to and whether they were previously documented or if you found new ones. I recommend the following wording, “Lidar documented in detail two known megasites (Cotoca and Landivar), as well as 24 smaller sites...” I have found myself using Figure 1 and Supplemental Table 1 to follow along with the geography of your study region, so think it would be helpful to have in parentheses the names for comparison. Also, are these the only megasites in this region, or are there others? You might consider adjusting Figure 1 to show which sites you confirmed using LiDAR, which were new sites, and also the location of megasites in the region.”

Response: Cotoca and Landivar are the only megasites known to us, but future research will undoubtedly offer us some more surprises. We have modified Fig. 1 so that the hierarchy of site sizes is colour-coded.

“Figure 2 and other Extended Data LiDAR figures: These figures (or at least a few) could be improved to label types of observed cultural features. At the moment, we see the morphology of various features in Figure 2, but to someone not familiar with the landscape in this region, it is not obvious what they are. Features are labeled in Extended Data Figure 1 with boxes and numbers, but the numbers are not described in the caption directly, so it’s not really made clear what the objects are. There are several figures in the Extended Data where features are numbered but not described in the caption or directly cited in the text. I do see some areas in the text where features are described and the figure is cited, but since there will likely be readers who are not familiar with this type of architecture or features, I would recommend ensuring all of your Extended Data captions make note of what the numbered/lettered cultural features are so it is clear to the reader.”

Response: Agreed. We have added information on the numbered cultural features to the Extended Data Figs. 1-4.

Extended Data Fig. 1 | Map of the Cotoca site with architectural features and location of the profile cuts of the three enclosures (see Extended Data Fig. 15). Numbered features: 1, principal, U-shaped mound of 22m height; 2-21 smaller platform mounds; 22 and 23, Platforms at the junction of a causeway and the polygonal enclosure.

Extended Data Fig. 2 | Lidar image of the Landívar site (see cross sections in Extended Data Fig. 14). Numbered features: 1-5 platform mounds; 6-11, causeways; 12, structure of unknown function.

Extended Data Fig. 3 | A, Location of lidar transect for the El Cerrito area. B, Canals (blue) and causeways (red) found in the area, some of them connecting the pre-Hispanic sites (red triangles). C. Lidar image of El Cerrito (see cross sections in Extended Data Fig. 14). Numbered features: 1, core area platform; 2, truncated pyramid, 3-5, platform mounds.

Extended Data Fig. 4 | Map of the Santa María site, located in the Cotoca area. The layout of the settlement on both sides of the paleo-river is unusual. As at the Salvatierra site (see Extended Data Fig. 5), water from the paleo-river was channeled into a circular pond located outside the walled enclosure. Numbered features: 1-4, platform mounds; 5 and 6, reservoir; 7 rampart and moat.

Referee #7: H. Clarity/Context

“Overall, the study is well-written and the structure works well. There do seem to be some areas where I found the text to read somewhat awkwardly. For example, the sentence in the abstract “Here we present lidar data from the Casarabe culture...” suggests that the LiDAR data itself comes from that culture. I would suggest changing the sentence to reflect that LiDAR data was collected in a

region of the world where that culture lived, as an example of one grammatical change. The title wording “Lidar evidence of megasites...” also does not seem correct, and I’d suggest considering something such as “Lidar reveals evidence of megasites...” “

Response: Agreed. We have substituted “*Here we present lidar data from the Casarabe culture...*” with “Here we present lidar data of settlement sites of the Casarabe culture”.

We hope that we have satisfactorily responded to the concerns of the reviewers and that our manuscript is now acceptable for publication in *Nature*.

Sincerely,

For the authors,
Heiko Prümers

Reviewer Reports on the Second Revision:

Referees' comments:

Referee #5 (Remarks to the Author):

The abstract refers to ancient agrarian low-density urbanism and refers to Asia, Africa and Central America.

My apologies that I did not comment on this previously.

1. It would not be appropriate to use the term "ancient" because, except for the very early periods of the Sri Lankan cases the settlements referred are more recent than the 5th century CE which is the convention for the end of the "Ancient".

The term "ancient" is not necessary and is inappropriate

Note Angkor has incorrectly been called "ancient" which is an English mistranslation of the French "ancien" which does not mean "ancient" eg the Ancien Regime in France was hardly "ancient".

2. The mention in the abstract of Africa and the reference given for that region apparently having agrarian, low-density urbanism should have caught my attention earlier. I think the paper by Dylan and Douglas is meant to be used as a source re-methodology elsewhere in the paper because so far as I can see that paper makes no reference to "agrarian low-density urbanism" in Africa. My apologies if I am mistaken.

If I am correct, referring to Africa in the opening sentence is incorrect.

What should be stated is that agrarian, low-density urbanism occurs in SE Asia (eg Angkor), and in Southern Asia - or refer to Sri Lanka - (see Coningham and Gunawardhana ref below), and in Central America (eg the Classic Maya).

Coningham, R.A.E. and Gunawardhana, P. (2013). Anuradhapura: Volume 3 The Hinterland. Oxford: BAR International Series.

Referee #7 (Remarks to the Author):

My thanks to the authors for their time and effort to address my comments. In my opinion they have adequately dealt with all of my concerns. A few minor items I noticed in the updated text for the title and lidar methods:

I suggest that the authors consider retaining the word "Lidar" somewhere in the title. While the study's main conclusions pertain to the settlement system, lidar is indeed the main methodological

focus of the study, and other researchers looking for papers about lidar in archaeological use would find the study more easily were that the case.

Line 65: You might consider spelling out the acronym lidar here upon first use in the text Lidar (Light Detection and Ranging)...

Line 293: It is more common for points which have been classified as “ground” to be used in a DEM, rather than those that are last return. Generally this is because in open areas, the first return can also be representative of the ground surface, and sometimes last return points can be representative of low vegetation, if the pulse did not make it all the way to the ground. Therefore ensuring the proper classification algorithms to discriminate ground from vegetation is important, and using classification rather than filtering is an ideal way of generating a DEM. In this case, I think given the very high density of the point cloud, fine resolution of the resulting DEMs, and that this seems to be a densely vegetated tropical area, this likely does not influence the outcome of the study, but I did want to mention it.

Line 296/297: ‘DSM’ is used on 296 and DEM following it on 297; recommend using one or the other and also spell out the acronym upon use.

Line 297: ‘las-files’ should be ‘LAS files’

Author Rebuttals to Second Revision:

In this letter, we explain the revisions we have made in response to comments by the referees #5 and #7. Each comment of the referee (in italics) is followed by our response (in normal text).

We thank Reviewers #5 and #7 for their helpful comments in relation to our revised manuscript.

Referee #5 / Use of the term megasite:

The abstract refers to ancient agrarian low-density urbanism and refers to Asia, Africa and Central America.

1. It would not be appropriate to use the term "ancient" because, except for the very early periods of the Sri Lankan cases the settlements referred are more recent than the 5th century CE which is the convention for the end of the "Ancient".

The term "ancient" is not necessary and is inappropriate

Note Angkor has incorrectly been called "ancient" which is an English mistranslation of the French "ancien" which does not mean "ancient" eg the Ancien Regime in France was hardly "ancient".

2. The mention in the abstract of Africa and the reference given for that region apparently having agrarian, low-density urbanism should have caught my attention earlier. I think the paper by Dylan and Douglas is meant to be used as a source re-methodology elsewhere in the paper because so far as I can see that paper makes no reference to "agrarian low-density urbanism" in Africa. My apologies if I am mistaken.

If I am correct, referring to Africa in the opening sentence is incorrect.

What should be stated is that agrarian, low-density urbanism occurs in SE Asia (eg Angkor), and in Southern Asia - or refer to Sri Lanka - (see Coningham and Gunawardhana ref below), and in Central America (eg the Classic Maya).

Coningham, R.A.E. and Gunawardhana, P. (2013). Anuradhapura: Volume 3 The Hinterland. Oxford: BAR International Series.

Response: Agreed. We have modified the first phrase of the abstract and we deleted the Ref on Africa, adding the reference Fletcher 2012 instead. The text now reads:

“Archaeological remains of agrarian-based, low-density urbanism¹⁻³ have been reported to exist beneath the tropical forests of SE Asia, Sri Lanka and Central America⁴⁻⁶.”

1. Fletcher, R. Low-Density, Agrarian-Based Urbanism: A Comparative View. *Insights* **2**, 2–19 (Durham University, 2009).
2. Fletcher, R. Low-Density, Agrarian-Based Urbanism: Scale, Power, and Ecology. in: *The comparative archaeology of complex societies* (ed. Michael E. Smith) 285-320 (Cambridge University Press, 2012).
3. Lucero, L. J., Fletcher, R. Coningham, R. From ‘collapse’ to urban diaspora: the transformation of low-density, dispersed agrarian urbanism. *Antiquity* **89**, 1139-1154 (2015).

4. Evans, D.H., Fletcher, R.J., Pottier, C., Chevance, J.-B., Soutif, D., Tan, B.S., Im, S., Ea, D., Tin, T., Kim, S. Uncovering archaeological landscapes at Angkor using lidar. *Proceedings of the National Academy of Sciences* 110, 12595-12600 (2013).
5. Canuto, M.A., Estrada-Belli, F., Garrison, T.G., Houston, S.D., Acuña, M.J., Kováč, M., Marken, D., Nondédéo, P., Auld-Thomas, L., Castanet, C. Ancient lowland Maya complexity as revealed by airborne laser scanning of northern Guatemala. *Science* 361 (2018).
6. Chase, A.F., Chase, D.Z., Fisher, C.T., Leisz, S.J., Weishampel, J.F. Geospatial revolution and remote sensing LiDAR in Mesoamerican archaeology. *Proceedings of the National Academy of Sciences* 109, 12916-12921 (2012).

Referee #7 / Lidar Methods:

I suggest that the authors consider retaining the word “Lidar” somewhere in the title. While the study’s main conclusions pertain to the settlement system, lidar is indeed the main methodological focus of the study, and other researchers looking for papers about lidar in archaeological use would find the study more easily were that the case.

Response: Agreed. The title now reads:

“Lidar reveals pre-Hispanic low-density urbanism in the Bolivian Amazon”

Line 65: You might consider spelling out the acronym lidar here upon first use in the text Lidar (Light Detection and Ranging)...

Response: Agreed. We have added “(Light Detection and Ranging)” in Line 64.

Line 293: It is more common for points which have been classified as “ground” to be used in a DEM, rather than those that are last return. Generally this is because in open areas, the first return can also be representative of the ground surface, and sometimes last return points can be representative of low vegetation, if the pulse did not make it all the way to the ground. Therefore ensuring the proper classification algorithms to discriminate ground from vegetation is important, and using classification rather than filtering is an ideal way of generating a DEM. In this case, I think given the very high density of the point cloud, fine resolution of the resulting DEMs, and that this seems to be a densely vegetated tropical area, this likely does not influence the outcome of the study, but I did want to mention it.

Response: We agree. Of course, different evaluation methods were tested. For our LIDAR data from Bolivia, the best results could be obtained in the way we used and describe.

Line 296/297: 'DSM' is used on 296 and DEM following it on 297; recommend using one or the other and also spell out the acronym upon use.

and

Line 297: 'las-files' should be 'LAS files'

Response: Agreed. The text was modified to “At the end of this process DEM LAS files were generated that had a mean point spacing of 0,3m.”

We hope that we have satisfactorily responded to the concerns of the reviewers and that our manuscript is now acceptable for publication in Nature.

Sincerely,

For the authors,
Heiko Prümers